# Winners and losers from Pfizer and Biontech's vaccine announcement: Evidence from S&P 500 (Sub)sector indices

**Burcu Kapar**[1]*, **Steven Buigut**[2], **Faisal Rana**[1]

**1** American University in Dubai, Dubai, United Arab Emirates, **2** Canadian University Dubai, Dubai, United Arab Emirates

* bkapar@aud.edu, burcu.kapar@gmail.com

## Abstract

This study explores how the US stock market reacted to the news of a successful development of vaccine by Pfizer and Biontech on November 9, 2020. In particular, the study analyses the effect of the vaccine announcement on 11 sector indices and 79 subsector indices. A key contribution of the present study is to provide a deeper subsector level of analysis lacking in existing literature. An event study approach is applied in identifying abnormal returns due to the November 9th vaccine announcement. Several event periods (-1, 0, 1, 2, 3, 0-1, 0-3) are analysed to provide a more complete picture of the effects. Based on analysis, it is established that there are considerable inter and intra sectoral variations in the impact of the vaccine news. The results show that the impact follows a clear pattern. The sectors that were hit hardest by the pandemic such as energy, financials, as well as subsectors like hotels and casinos, benefited the most from positive vaccine news. Subsectors that gained from the pandemic such as airfreight, household appliances and computers and electronics retail were depressed the most by the news. These findings suggest that while the availability of vaccines is expected to help steer economies gradually to normalcy, the re-adjustment is likely to be asymmetric across subsectors. While some subsectors expect to expand as these industries recover from the contraction inflicted by the COVID-19 environment, other subsectors expect adjustment losses as these industries shed off the above average gains driven by the COVID-19 environment.

## 1 Introduction

The COVID-19 pandemic has affected countries and businesses in many different ways. The implementation of government restrictions on commercial activity and social distancing measures have caused extreme volatility in global financial markets that is unmatched in recent decades [1–3]. For example, the Dow Jones Industrial Average (DIJA) index lost more than a quarter of its value in only four trading days in March 2020 [4]. Although markets picked up and share prices rose to even above the pre-pandemic levels, the recovery in the US stock market was mainly confined to stocks of technology and tech-enabled companies such as

**Data Availability Statement:** Data from article "Winners and Losers from Pfizer and Biontech's Vaccine Announcement: Evidence from S&P 500 (Sub)Sector Indices" are within the Supporting information files. The data are also in the

Qualitative Data Repository (https://doi.org/10.5064/F64FB61X). QDR Main Collection (view at https://data.qdr.syr.edu/dataverse/main).

**Funding:** The author(s) received no specific funding for this work.

**Competing interests:** The authors have declared that no competing interests exist.

Facebook, Apple, Amazon, Netflix, and Google, that witnessed a considerable surge in demand due to lockdown policies [5].

As expected, stock markets showed some improvement on the announcement of success in phase-3 study of Pfizer and Biontech vaccine candidate against COVID-19 on November 9, 2020. The study indicated that the vaccine candidate reduced the risk of infection by around 90 percent [6]. However, there are striking sectoral disparities on the impact of the vaccine news on the share prices of various sectors [4, 5]. According to [7], the spread between the best and worst-performing sectors increased from 27 percentage points in mid-March to 80 percentage points over the year.

It is widely accepted that share prices reflect available information and adjust swiftly to current news and events [8, 9], though other studies acknowledge (some) markets are less than fully efficient [10–12]. Understanding how different sectors and subsectors are impacted by the news will help investors and policy makers to formulate optimal responses. This requires an in-depth analysis at both sector and subsector levels to examine the variation in the impact of the news on share prices. However, the focus of existing literature on the impact of vaccine development [13–15] is on the aggregate market level, not on subsector impact.

The current study delves deeper to provide a better understanding of the impact of vaccine development news. The aim of the study is twofold. First it examines the effect of the Pfizer and Biontech November 9, 2020, announcement. Thus, it contributes to the growing body of research on the impact of the COVID-19 pandemic on financial markets by exploring the COVID-19 vaccines' impact on stock market. The second objective of the current research is to disaggregate the analysis to both sectoral and sub-sectoral levels. By focusing on sectoral/sub-sectoral level, this paper expands the existing literature that currently is mainly focused on aggregate equity markets [1, 16–24] and therefore ignore the heterogenous effects at the sub-sector level.

The results of the study suggest that the announcement generates optimism in a wide range of sectors and subsectors, though the impact has been disparate. Interestingly while some sectors exhibit no significant immediate effect overall, the sub-sectors do show some (sometimes diverse) effects. The findings from this study reveal the impact of the vaccine announcement that is opposite to that exhibited by the market at the onset of the COVID-19 pandemic found by [4].

The rest of this study is structured as follows. Section 2 discusses the literature review; Section 3 presents the data; Section 4 explains the methodology; Section 5 presents empirical results and discussion; and Section 6 concludes and presents policy recommendations.

## 2 Literature review

With the outbreak of the coronavirus (COVID-19) pandemic, an increasing number of researchers have examined the impact of the pandemic on stock markets. [1] document that no previous infectious disease outbreak, including the Spanish Flu, has impacted the stock market as forcefully as the COVID-19 pandemic due to strict government restrictions on commercial activity and voluntary social distancing. Exploring the direct effects and spillovers of COVID-19, [19] find that COVID-19 has a negative but short-term impact on stock markets of affected countries. By using a large sample of 63 stock markets covering all key markets, [16] find that the Wuhan lockdown induces negative spillover effects on markets in Europe, North America and other global markets. This is mainly attributed to fear and uncertainty as these markets had yet to introduce domestic restrictions and had minimal infections at the time. The rapid transmission of cases outside China particularly in Europe and the

introduction of containment measures result in severe market decline which highlights the need for quick, globally coordinated response to contagious diseases.

Controlling for traditional market drivers (such as investor sentiment, credit risk, liquidity risk, safe-haven asset demand and the price of oil), [23] conclude that the daily total count of confirmed COVID-19 cases is a leading factor in influencing equity prices. Using panel data analysis, [25] estimate that both the daily growth in confirmed cases and number of deaths caused by COVID-19 have significant negative effects on returns in Chinese stock market. [26] finds that stock markets react more strongly to the growth in number of confirmed cases as compared to the growth in number of deaths. [4] examine the US stock market during the crash of March 2020. They estimate that approximately 90% of the S&P 1500 stocks generate asymmetrically distributed large negative returns. The consensus from this emerging literature on COVID-19 suggests that stock markets respond negatively and significantly to COVID-19. Individual stock responses may vary, however, depending on several factors.

There are few studies documenting the role of vaccine development on stock markets. [15] using data for 66 markets, show that mass vaccinations significantly decreased the stock market volatility, even after controlling for both the pandemic and government policy interventions. [27] examine the volatility of the energy stocks in fifty-eight countries. They document that vaccination programs assist in decreasing the volatility of energy stocks around the world. Interestingly both [15, 27] find that the stabilizing effect of vaccination on volatility is more pronounced in developed markets than in emerging ones. [28] study the role of the vaccine initiation rate in mitigating the stock market volatility on a global scale. They highlight the positive effect of the vaccine initiation rate in stabilizing the international stock markets. Consistent with earlier studies, they also find a stronger impact of vaccinations in developed markets.

[14] focus on the impact of vaccine news announcements by leading vaccine companies on the financial and commodity markets from January to December 2020. They show that the vaccine announcements of Johnson and Johnson, Moderna, Oxford-AstraZeneca, and Pfizer-BioNTech have impacted the stock markets in the U.S. and Europe, while stocks markets in Asia and Australia are unaffected. The impact is also varied in the commodities markets. Transportation commodities (crude oil and gasoline) react to the announcements while metal and construction commodities (gold, copper, and lumber) remain unmoved. Overall, they conclude that the effect of the announcements is statistically significant on the stock prices, interest rates, commodity currencies, and transportation commodities.

In another study, [29] show that global stock markets react positively to different phases of human clinical trials for COVID-19 vaccine candidates. After controlling for both the pandemic itself and investor sentiment, they find a positive abnormal return in forty-nine countries on the first day of clinical trials. Phase I and Phase II of the human clinical trials of vaccine candidates are more effective on the stock market.

[13] develop an asset-pricing model to forecast the value of medicine by employing a vaccine progress indicator. They estimate that the stock market improves by around 8.6 percent if the expected time to vaccine development decreases by a year. [30] develops a model of pandemic risk management that is integrated into an asset-pricing structure. They conclude that the quick arrival of vaccines lowers the duration of the pandemic and minimizes the impact of shocks.

All the studies mentioned have focused on the effect of vaccine announcement on the aggregate stock market. To the best of the author's knowledge, the effect of vaccinations on the stock prices at the sectoral and sub-sectoral level remain unexplored. Some studies such as [31, 32] have analysed the energy companies, but their interest is on the impact of fiscal pressure.

## 3 Data

This study covers the daily stock prices of 11 sector and 79 sub-sector indices of the S&P 500. The 11 sectors, as per the Global Industry Classification Standard (GICS), are as follows: Information Technology, Health Care, Financials, Consumer Discretionary, Communication Services, Industrials, Consumer Staples, Energy, Utilities, Real Estate, and Materials. Data is obtained from Thomson Reuters Eikon for the period 23 June 2020 to 12 November 2020, for a total of 139 daily observations per index. This period includes the announcement date, on November 9, 2020, by Pfizer and Biontech that their vaccine candidate was successful in Phase-3. The S&P 500 index measures the stock performance of 500 large companies listed on stock exchanges in the U.S. and is widely used as an indicator of the performance of global equities. For each index, daily return is calculated as the natural logarithmic first difference of the daily closing price multiplied by 100.

Table 1 reports the descriptive statisticsal of the sectoral indices considered in this study over the period from 23 June 2020 to 12 November 2020. According to the table, all energy indices, biotechnology from health care sector, food retail and drug retail from consumer staples sector, internet services and infrastructure from information technology sector and gas utilities from utilities sector give negative average return for investors. Most of the series are moderately skewed having positive or negative skewness. Majority of the indices have high kurtosis which indicates fat tails in the distributions.

## 4 Methodology

Event study methodology is one of the most frequently used analytical tools in financial research. The objective of an event study is to assess whether there are any abnormal or excess returns earned by security holders accompanying specific events (e.g., earnings announcements, merger announcements, stock splits) where an abnormal or excess return is the difference between observed return and that expected return given a particular return generating model. In this study, we have exploited market-model event-study methodology to identify abnormal returns resulting from the announcement of the vaccine.

### 4.1 Event study methodology

The market model is the most frequently used expected return model. For any stock market index i, the market model can be presented as Eq 1:

$$R_{it} = \hat{\alpha}_i + \hat{\beta}_i R_{mt} + \epsilon_{it} \tag{1}$$

where $R_{it}$ represents return of a sectoral or sub-industry indices on day t which belongs to estimation window, $R_{mt}$ denotes the return of the S&P 500 Index on day t belonging to the same period. $\hat{\alpha}_i$ and $\hat{\beta}_i$ are the parameters of the market model. The expected return $E(Ri)$ is then calculated as in Eq 2 and $AR_{it}$ which represents the abnormal return of any sectoral or sub-industry indices on day t is determined from Eq 3.

$$E(R_{it}) = \hat{\alpha}_i + \hat{\beta}_i R_{mt} \tag{2}$$

while;

$$AR_{it} = R_{it} - E(R_{it}) \tag{3}$$

To measure the total impact of an event right after the announcement, the "cumulative abnormal return (CAR)" for the event window of [0, 1] and [0, 3] is calculated as follows in

**Table 1. Descriptive statistics.**

| Index | Mean | Standard Deviation | Minimum | Maximum | Skewness | Kurtosis | Jarque-Bera Test Statistics |
|---|---|---|---|---|---|---|---|
| **Financials** | 0.11 | 1.66 | -4.43 | 7.85 | 0.71 | 6.71 | 66.03 |
| Banks | 0.07 | 2.39 | -6.32 | 12.48 | 1.18 | 8.77 | 162.2 |
| Insurance | 0.10 | 1.46 | -3.22 | 5.48 | 0.29 | 3.72 | 3.55 |
| Capital Markets | 0.07 | 1.35 | -3.88 | 3.14 | -0.52 | 3.33 | 5.10 |
| Consumer Finance | 0.16 | 2.74 | -6.16 | 15.12 | 1.41 | 10.51 | 268.1 |
| Diversified Financial Services | 0.22 | 1.28 | -3.03 | 5.87 | 0.62 | 6.07 | 45.75 |
| **Energy** | -0.20 | 2.60 | -5.7 | 13.3 | 1.37 | 8.87 | 175 |
| Oil& Gas Exploration and Production | -0.17 | 3.25 | -7.40 | 15.13 | 0.89 | 6.57 | 66.34 |
| Oil & Gas Equipment Services | -0.07 | 3.63 | -8.23 | 17.01 | 0.87 | 6.69 | 69.54 |
| Oil & Gas Drilling | -0.35 | 4.50 | -11.23 | 16.05 | 0.18 | 3.96 | 4.43 |
| Energy Equipment and Services | -0.04 | 3.44 | -8.87 | 17.14 | 0.88 | 8.17 | 124.8 |
| **Real Estate** | 0.04 | 1.31 | -2.95 | 2.54 | -0.17 | 2.32 | 2.44 |
| Equity Real Estate Investment Trusts | 0.04 | 1.30 | -2.97 | 2.61 | -0.17 | 2.31 | 2.498 |
| Real Estate Mng.and Dev. | 0.18 | 2.86 | -7,34 | 15.53 | 1.84 | 11.61 | 365.5 |
| **Communication Services** | 0.12 | 1.52 | -4.59 | 4.16 | -0.44 | 3.75 | 5.56 |
| Wireless Telecommunication Services | 0.17 | 1.68 | -4.63 | 6.27 | 0.27 | 5.13 | 20.04 |
| Interactive Media and Services | 0.16 | 2.12 | -6.89 | 6.63 | -0.32 | 4.18 | 7.60 |
| Broadcasting | 0.04 | 2.15 | -4.68 | 4.29 | -0.26 | 2.52 | 2.12 |
| Interactive Home Entertainment | -0.01 | 1.83 | -4.43 | 3.75 | -0.44 | 2.98 | 3.33 |
| Media and Entertainment | 0.14 | 1.75 | -5.05 | 5.07 | -0.31 | 3.69 | 3.57 |
| Integrated Telecommunication Services | 0.02 | 0.98 | -2.84 | 3.13 | 0.28 | 4.51 | 10.87 |
| Diversified Telecommunication Services | 0.02 | 0.99 | -2.82 | 3.07 | 0.26 | 4.34 | 8.64 |
| **Health Care** | 0.10 | 1.14 | -3.28 | 4.35 | 0.14 | 4.57 | 10.58 |
| Biotechnology | -0.08 | 1.56 | -3.87 | 7.46 | 0.97 | 7.33 | 93.81 |
| Health Care Equipment and Supplies | 0.16 | 1.29 | -4.02 | 2.85 | -0.31 | 3.68 | 3.57 |
| Health Care Distributors | 0.08 | 1.77 | -3.42 | 5.28 | 0.42 | 2.99 | 2.94 |
| Heath Care Facility | 0.35 | 2.54 | -6.03 | 10.12 | 0.50 | 4.75 | 16.97 |
| Health Care Technology | 0.04 | 1.48 | -3.43 | 3.74 | -0.03 | 3.07 | 0.04 |
| Life Sciences Tools and Services | 0.21 | 1.58 | -5.01 | 3.51 | -0,57 | 3.75 | 7.83 |
| Pharmaceuticals | 0.05 | 1.07 | -3.01 | 4.16 | 0.11 | 4.53 | 9.98 |
| **Consumer Discretionary** | 0.15 | 1.47 | -3.63 | 3.16 | -0.26 | 2.67 | 1.547 |
| Auto Components | 0.25 | 2.15 | -5.69 | 4.89 | -0.11 | 2.80 | 0.35 |
| Automobiles | 0.32 | 2.10 | -4.93 | 4.93 | 0.004 | 2.63 | 0.57 |
| Hotels | 0.19 | 3.35 | -7.29 | 17.03 | 1.43 | 9.05 | 187 |
| Casinos and Gaming | 0.16 | 3.23 | -7.24 | 13.90 | 1.08 | 5.78 | 51.86 |
| Restaurants | 0.17 | 1.14 | -3.69 | 2.78 | -0.45 | 3.43 | 4.29 |
| Household Durables | 0.21 | 1.94 | -4.94 | 4.22 | -0.17 | 2.73 | 0.81 |
| Home Building | 0.21 | 2.48 | -7.10 | 6.51 | -0.02 | 3.06 | 0.02 |
| Household Appliances | 0.36 | 2.23 | -10.96 | 7.69 | -0.91 | 9.18 | 173.4 |
| Consumer Electronics | 0.15 | 1.48 | -4.34 | 5.49 | 0.20 | 4.45 | 9.54 |
| Internet and Direct Marketing Retail | 0.12 | 2.41 | -5.31 | 6.94 | 0.25 | 2.90 | 1.13 |
| Multiline Retail | 0.18 | 1.29 | -3.81 | 6.19 | 0.53 | 7.03 | 72.41 |
| Distributors | 0.15 | 1.88 | -6.10 | 5.05 | -0.31 | 3.96 | 5.53 |
| General Merchandise Stores | 0.18 | 1.31 | -3.81 | 6.36 | 0.60 | 7.40 | 86.7 |
| Specialty Retail | 0.12 | 1.26 | -3.77 | 2.66 | -0.39 | 2.97 | 2.65 |
| Computers and Electronics Retail | 0.28 | 2.05 | -10.31 | 7.55 | -1.19 | 10.12 | 235.2 |
| Home Furnishing Retail | 0.12 | 2.87 | -15.05 | 7.61 | -1.31 | 9.69 | 214.9 |

*(Continued)*

**Table 1.** (Continued)

| Index | Mean | Standard Deviation | Minimum | Maximum | Skewness | Kurtosis | Jarque-Bera Test Statistics |
|---|---|---|---|---|---|---|---|
| Textiles, Apparel and Luxury Goods | 0.22 | 1.81 | -7.02 | 6.47 | -0.31 | 5.71 | 32.09 |
| **Consumer Staples** | 0.12 | 0.91 | -2.96 | 1.97 | -0.51 | 3.79 | 7.13 |
| Food and Staples Retailing | 0.17 | 1.11 | -2.57 | 3.12 | 0.05 | 3.11 | 0.10 |
| Drug Retail | -0.05 | 2.31 | -8.07 | 6.42 | 0.12 | 4.04 | 4.74 |
| Food Retail | -0.002 | 1.62 | -6.73 | 4.34 | -0.80 | 5.67 | 40.52 |
| Hypermarkets and Super Centers | 0.21 | 1.26 | -3.33 | 4.32 | 0.23 | 4.19 | 6.81 |
| Brewers | 0.11 | 2.21 | -5.25 | 7.15 | 0.43 | 3.68 | 5.00 |
| Soft Drinks | 0.12 | 1.16 | -3.80 | 3.40 | -0.25 | 3.97 | 4.96 |
| Food Products | 0.06 | 1.08 | -3.33 | 2.41 | -0.44 | 3.55 | 4.59 |
| Agricultural Products | 0.22 | 0.98 | -2.35 | 3.38 | 0.29 | 3.62 | 2.98 |
| Packaged Foods and Meats | 0.05 | 1.09 | -3.38 | 2.47 | -0.49 | 3.53 | 5.33 |
| Household Products | 0.15 | 0.99 | -3.82 | 1.87 | -0.98 | 5.43 | 40.85 |
| Personal Products | 0.24 | 1.62 | -6.80 | 4.20 | -0.81 | 6.30 | 56.48 |
| Tobacco | 0.016 | 1.43 | -4.16 | 4.05 | 0.08 | 4.09 | 5.06 |
| **Industrials** | 0.19 | 1.44 | -3.58 | 3.29 | -0.35 | 3.06 | 2.12 |
| Airlines | 0.11 | 3.42 | -7.97 | 13.73 | 0.79 | 5.24 | 31.65 |
| Railroads | 0.22 | 1.55 | -3.06 | 3.35 | -0.32 | 2.18 | 4.48 |
| Transportation Infrastructure | 0.18 | 1.74 | -4.26 | 7.03 | 0.35 | 4.91 | 17.33 |
| Air Freight and Logistics | 0.43 | 1.93 | -7.94 | 8.02 | -0.02 | 8.10 | 108.8 |
| Building products | 0.31 | 1.56 | -4.09 | 4.03 | -0.31 | 3.58 | 3.15 |
| Aerospace and Defense | 0.008 | 1.98 | -4.35 | 6.81 | 0.40 | 3.87 | 5.87 |
| Electrical Equipment | 0.21 | 1.67 | -4.94 | 4.58 | -0.21 | 3.39 | 1.38 |
| Industrial Conglomerates | 0.18 | 1.68 | -4.65 | 4.35 | -0.42 | 3.29 | 3.27 |
| Machinery | 0.28 | 1.55 | -4.03 | 3.85 | -0.34 | 3.14 | 2.09 |
| **Information Technology** | 0.14 | 1.77 | -6.00 | 3.76 | -0.71 | 3.78 | 11.02 |
| Communications Equipment | -0.08 | 1.59 | -9.32 | 2.86 | -2.13 | 13.00 | 492.5 |
| IT Services | 0.06 | 1.45 | -4.41 | 3.49 | -0.54 | 3.43 | 5.75 |
| IT Consulting and Other Services | 0.09 | 1.39 | -3.78 | 4.62 | -0.19 | 4.35 | 8.20 |
| Data Processing and Outsourcing | 0.06 | 1.56 | -4.63 | 4.15 | -0.42 | 3.38 | 3.54 |
| Internet Services and Infrastructure | -0.034 | 1.72 | -6.75 | 5.02 | -0.80 | 5.61 | 39.16 |
| Semiconductors and Equipment | 0.18 | 1.92 | -6.42 | 3.96 | -0.49 | 3.58 | 5.57 |
| Software | 0.09 | 2.02 | -5.89 | 4.73 | -0.33 | 3.32 | 2.34 |
| Technology Hardware and Storage | 0.25 | 2.62 | -8.15 | 9.62 | -0.12 | 4.70 | 12.33 |
| **Materials** | 0.18 | 1.47 | -3.47 | 3.97 | -0.22 | 2.91 | 0.87 |
| Chemicals | 0.16 | 1.54 | -3.44 | 4.25 | -0.11 | 2.88 | 0.25 |
| Construction Materials | 0.18 | 2.14 | -9.09 | 5.22 | -0.84 | 6.01 | 49.5 |
| Containers and Packaging | 0.25 | 1.49 | -3.11 | 3.26 | -0.15 | 2.62 | 0.98 |
| Metals and Mining | 0.22 | 1.94 | -5.58 | 4.72 | -0.46 | 3.72 | 5.78 |
| **Utilities** | 0.14 | 1.14 | -2.98 | 3.08 | -0.03 | 2.63 | 0.59 |
| Electric Utilities | 0.14 | 1.19 | -2.78 | 3.34 | 0.08 | 2.50 | 1.17 |
| Gas Utilities | -0.001 | 1.39 | -4.62 | 4.65 | 0.46 | 5.19 | 23.67 |
| Water Utilities | 0.25 | 1.30 | -4.59 | 3.13 | -0.24 | 3.72 | 3.19 |
| Multi Utilities | 0.11 | 1.21 | -3.42 | 2.74 | -0.24 | 3.09 | 0.99 |

This table reports descriptive statistics of indices. Data is obtained from Thomson Reuters Eikon for the period from 23 June 2020 to 12 November 2020, for a total of 139 daily observations per index.

Eq 4:

$$CAR_{i,(t_1,t_2)} = \sum_{t=t_1}^{t_2} AR_{i,t} \tag{4}$$

where $t_1$ and $t_2$ represent the start and end of event window.

Finally, after identifying all abnormal returns and cumulative abnormal returns over chosen event windows, their statistical significance are tested with t-test. The null and alternative hypotheses are stated as in Eq 5. The critical value for the null hypothesis rejection is ± 2.576, 1.96 and 1.645 with the confidence level of 99%, 95% and 90%, respectively.

$$H_0 : \overline{AR_t} = 0, H_1 = \overline{AR_t} \neq 0$$
$$H_0 : \overline{CAR_t} = 0, H_1 = \overline{CAR_t} \neq 0 \tag{5}$$

Fig 1 below presents the event timeline. Estimation window of 90 days dating from 23 June 2020 to 28 October 2020 is selected. The main reason for choosing a short estimation window is to exclude the early period of the pandemic where S&P 500 experienced high volatility. In case there is an information leakage in the market before the announcement, the data dating from 29 October 2020 to 6 November 2020, i.e. 7 days preceding the event from the estimation window is excluded (see Fig 1 below). Six different event windows: [-1], [0], [1], [2], [3], [0, 1] and [0, 3] are considered to measure the instantaneous effect of the announcement.

## 5 Empirical results and discussion

Uncertainty and fear have been a feature of the COVID-19 pandemic environment especially during the early period. The uncertainty and non-pharmaceutical interventions such as social distancing, travel restrictions and lockdowns to limit the spread were the key factors driving the socio-economic impacts of the pandemic. There are several channels through which a vaccine announcement, even prior to its actual rollout, would benefit the economy. An affirmation that a vaccine was imminent would reduce the uncertainty and fear, and generally boost optimism in markets as economic actors anticipate a rollback of restrictive interventions.

The basic results (Table 2) show the vaccine announcement on November 9, 2020, has disparate effects on the market. The (sub)sectors that were hardest hit by the pandemic [4, 5, 33] benefit the most from positive vaccine news, while sectors that gained from the pandemic were depressed by the news. Financials and energy sectors exhibit clear positive effects on [0], [0, 1] and [0, 3]. The financial sector and subsectors show significant gains with the largest gains registered by consumer finance and banks at 13.94 percent and 11.46 percent on [0], respectively.

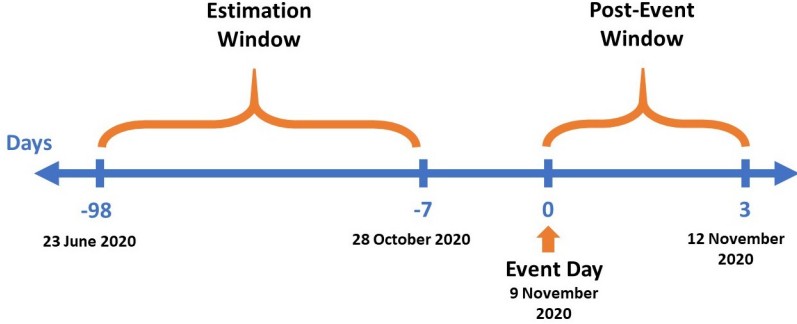

**Fig 1. Event timeline.**

**Table 2. Market model abnormal return and cumulative abnormal return for different event windows-estimation period: 23 June 2020–28 October 2020.**

| Event Windows | [-1] | | [0] | | [1] | | [2] | | [3] | | [0, 1] | | [0, 3] | |
|---|---|---|---|---|---|---|---|---|---|---|---|---|---|---|
| Indices | AR | T-Stats. | AR | T-Stats. | AR | T-Stats. | AR | T-Stats. | AR | T-Stats. | CAR | T-Stats. | CAR | T-Stats. |
| **Financials** | -0.73% | -0.70 | 6.91% | 6.58*** | 0.90% | 0.86 | -1.10% | -1.05 | -0.77% | -0.73 | 7.81% | 5.2*** | 5.94% | 2.83*** |
| Banks | -1.66% | -0.97 | 11.46% | 6.70*** | 0.09% | 0.05 | -1.56% | -0.91 | -1.07% | -0.63 | 11.55% | 4.77*** | 8.92% | 2.61*** |
| Insurance | 0.12% | 0.12 | 4.63% | 4.58*** | 2.28% | 2.26** | -1.49% | -1.48 | -0.54% | -0.53 | 6.91% | 4.84*** | 4.88% | 2.41** |
| Capital Markets | -0.74% | -1.07 | 2.12% | 3.07*** | 0.07% | 0.10 | 0.34% | 0.49 | -0.89% | -1.29 | 2.19% | 2.24*** | 1.64% | 1.19 |
| Consumer Finance | -1.31% | -0.68 | 13.94% | 7.19*** | -0.51% | -0.26 | -3.92% | -2.02** | 0.53% | 0.27 | 13.43% | 4.90*** | 10.04% | 2.59*** |
| Diversified Financial Services | 0.62% | 0.75 | 5.02% | 6.05*** | 3.12% | 3.76*** | -1.07% | -1.29 | -0.58% | -0.70 | 8.14% | 6.93*** | 6.49% | 3.91*** |
| **Energy** | -1.71% | -0.92 | 12.45% | 6.73*** | 3.07% | 1.66* | -1.25% | -0.68 | -1.93% | -1.04 | 15.52% | 5.93*** | 12.34% | 3.34*** |
| Oil & Gas Exploration and Production | -2.63% | -1.11 | 13.87% | 5.85*** | 3.55% | 1.50 | -1.42% | -0.60 | -2.33% | -0.98 | 17.42% | 5.20*** | 13.67% | 2.88*** |
| Oil & Gas Equipment Services | -1.82% | -0.66 | 15.70% | 5.67*** | 1.18% | 0.43 | -2.34% | -0.84 | -1.19% | -0.43 | 16.88% | 4.31*** | 13.35% | 2.41** |
| Oil & Gas Drilling | -1.89% | -0.49 | 14.80% | 3.86*** | 0.93% | 0.24 | -1.76% | -0.46 | -2.56% | -0.67 | 15.73% | 2.90*** | 11.41% | 1.49 |
| Energy Equipment and Services | -1.66% | -0.64 | 15.92% | 6.17*** | 0.31% | 0.12 | -2.92% | -1.13 | -0.38% | -0.15 | 16.23% | 4.45*** | 12.93% | 2.51** |
| **Real Estate** | -0.33% | -0.35 | 1.71% | 1.84* | 0.69% | 0.74 | 0.25% | 0.27 | -0.18% | -0.19 | 2.40% | 1.82* | 2.47% | 1.33 |
| Equity Real Estate Investment Trusts | -0.31% | -0.33 | 1.52% | 1.62 | 0.78% | 0.83 | 0.29% | 0.31 | -0.17% | -0.18 | 2.30% | 1.73* | 2.42% | 1.29 |
| Real Estate Mng.and Dev. | -1.26% | -0.53 | 9.32% | 3.92*** | -2.72% | -1.14 | -1.29% | -0.54 | -0.79% | -0.33 | 6.60% | 1.96** | 4.52% | 0.95 |
| **Communication Services** | -0.06% | -0.08 | -1.54% | -2.00** | -0.17% | -0.22 | -0.28% | -0.36 | 0.62% | 0.81 | -1.71% | -1.57 | -1.37% | -0.89 |
| Wireless Telecommunication Services | 5.29% | 4.34*** | -1.60% | -1.31 | -0.72% | -0.59 | 2.16% | 1.77 | 2.20% | 1.80 | -2.32% | -1.34 | 2.04% | 0.84 |
| Interactive Media and Services | -0.26% | -0.21 | -3.62% | -2.92*** | -1.54% | -1.24 | -0.12% | -0.10 | 1.04% | 0.84 | -2.32% | -1.34 | -4.24% | -1.71 |
| Broadcasting | -4.23% | -2.32** | 0.78% | 0.43 | 3.11% | 1.71* | -0.15% | -0.08 | -3.54% | -1.95** | 3.89% | 1.51 | 0.20% | 0.05 |
| Interactive Home Entertainment | -2.44% | -1.68* | -5.37% | -3.70*** | -0.06% | -0.04 | 1.30% | 0.90 | 0.73% | 0.50 | -5.43% | -2.65*** | -3.40% | -1.17 |
| Media and Entertainment | -0.26% | -0.28 | -2.17% | -2.33** | -0.57% | -0.61 | -0.30% | -0.32 | 0.71% | 0.76 | -2.74% | -2.08** | -2.33% | -1.25 |
| Integrated Telecommunication Services | 0.33% | 0.42 | 2.18% | 2.79*** | 2.14% | 2.74*** | -0.53% | -0.68 | -0.14% | -0.18 | 4.32% | 3.92*** | 3.65% | 2.34** |
| Diversified Telecommunication Services | 0.28% | 0.35 | 2.15% | 2.72*** | 2.22% | 2.81*** | -0.55% | -0.70 | -0.13% | -0.16 | 4.37% | 3.91*** | 3.69% | 2.34** |
| **Health Care** | 0.21% | 0.34 | -0.11% | -0.18 | 0.49% | 0.79 | -0.70% | -1.13 | 0.38% | 0.61 | 0.38% | 0.43 | 0.06% | 0.05 |
| Biotechnology | -0.36% | -0.34 | -2.31% | -2.20** | 2.14% | 2.04** | -0.34% | -0.32 | 0.18% | 0.17 | -0.17% | -0.11 | -0.33% | -0.16 |
| Health Care Equipment and Supplies | 0.62% | 0.64 | 0.92% | 0.95 | 0.67% | 0.69 | -1.00% | -1.03 | 0.22% | 0.23 | 1.59% | 1.16 | 0.81% | 0.42 |
| Health Care Distributors | -0.34% | -0.24 | 4.06% | 2.92*** | 2.95% | 2.12** | -3.64% | -2.62*** | 0.48% | 0.35 | 7.01% | 3.56*** | 4.41% | 0.96 |
| Heath Care Facility | 0.19% | 0.09 | 5.40% | 2.52** | 3.40% | 1.59 | -4.85% | -2.27** | -0.16% | -0.07 | 8.80% | 2.91** | 3.74% | 0.53 |
| Health Care Technology | 0.86% | 0.76 | 0.30% | 0.27 | 1.33% | 1.18 | -0.74% | -0.65 | -0.08% | -0.07 | 1.63% | 1.02 | 0.81% | 0.36 |
| Life Sciences Tools and Services | 1.93% | 2.03** | -6.20% | -6.53*** | -3.19% | -3.36*** | 1.21% | 1.27 | 1.47% | 1.55 | -9.39% | -6.99*** | -6.71% | -3.53*** |
| Pharmaceuticals | 0.34% | 0.33 | 1.36% | 1.31 | 0.79% | 0.76 | -0.91% | -0.88 | -0.24% | -0.23 | 2.28% | 2.37** | 1.32% | 0.38 |
| **Consumer Discretionary** | -0.32% | -0.49 | -2.91% | -4.48*** | -1.05% | -1.62 | 0.61% | 0.94 | -0.21% | -0.32 | -3.96% | -4.31*** | -3.56% | -2.74*** |
| Auto Components | -0.69% | -0.41 | 1.24% | 0.74 | 2.05% | 1.23 | 0.43% | 0.26 | -2.03% | -1.22 | 3.29% | 1.39 | 1.69% | 0.51 |
| Automobiles | -0.70% | -0.40 | 2.59% | 1.47 | 3.34% | 1.90* | -1.12% | -0.63 | -1.96% | -1.11 | 5.93% | 2.38** | 2.85% | 0.81 |
| Hotels | 2.87% | 1.01 | 6.74% | 2.37** | 10.04% | 3.54*** | -2.05% | -0.72 | -1.90% | -0.67 | 16.78% | 4.18*** | 23.97% | 2.54** |
| Casinos and Gaming | 0.64% | 0.23 | 12.77% | 4.64*** | -1.16% | -0.42 | -4.25% | -1.55 | -1.82% | -0.66 | 11.61% | 2.99*** | 5.54% | 1.01 |
| Restaurants | 0.04% | 0.05 | 0.33% | 0.43 | -1.01% | -1.33 | 0.96% | 1.26 | -1.44% | -1.89 | -0.68% | -0.63 | -0.51% | -0.20 |
| Household Durables | -2.26% | -1.60 | -6.17% | -4.38*** | 4.01% | 2.84*** | 0.23% | 0.16 | -1.23% | -0.87 | -2.16% | -1.08 | -3.16% | -1.12 |
| Home Building | -3.64% | -1.94* | -8.41% | -4.47*** | 6.51% | 3.46*** | 0.96% | 0.51 | -1.53% | -0.81 | -1.90% | -0.71 | -9.21% | -2.00** |

*(Continued)*

**Table 2.** (Continued)

| Event Windows | [-1] | | [0] | | [1] | | [2] | | [3] | | [0, 1] | | [0, 3] | |
|---|---|---|---|---|---|---|---|---|---|---|---|---|---|---|
| Indices | AR | T-Stats. | AR | T-Stats. | AR | T-Stats. | AR | T-Stats. | AR | T-Stats. | CAR | T-Stats. | CAR | T-Stats. |
| Household Appliances | 1.10% | 0.67 | -12.37% | -7.59*** | 1.06% | 0.05 | 0.08% | 0.05 | -0.12% | -0.07 | -11.31% | -4.91*** | -11.35% | -3.48*** |
| Consumer Electronics | 1.72% | 1.77 | -0.88% | -0.91 | -0.82% | -0.85 | -0.06% | -0.06 | -0.08% | -0.08 | -1.70% | -1.24 | -1.84% | -0.95 |
| Internet and Direct Marketing Retail | -0.20% | -0.12 | -5.52% | -3.41*** | -3.50% | -2.16** | 1.98% | 1.22 | 0.31% | 0.19 | -9.02% | -3.94*** | -0.95% | -2.08** |
| Multiline Retail | 1.39% | 1.26 | -4.55% | -4.14*** | 0.83% | 0.75 | 0.38% | 0.35 | 0.02% | 0.02 | -3.72% | -2.39** | -3.32% | -1.51 |
| Distributors | -1.37% | -0.94 | -4.99% | -3.42*** | 3.57% | 2.45** | -1.81% | -1.24 | -1.24% | -0.85 | -1.42% | -0.69 | -4.47% | -1.53 |
| General Merchandise Stores | 1.38% | 1.23 | -4.54% | -4.05*** | 0.83% | 0.74 | 0.38% | 0.34 | 0.01% | 0.01 | -3.71% | -2.34** | -3.32% | -1.48 |
| Specialty Retail | -0.53% | -0.71 | -3.09% | -4.12*** | 1.81% | 2.41** | -0.59% | -0.79 | 0.09% | 0.12 | -1.28% | -1.21 | -1.78% | -1.19 |
| Computers and Electronics Retail | -0.22% | -0.15 | -11.55% | -7.97*** | 1.33% | 0.92 | 0.80% | 0.55 | 0.12% | 0.08 | -10.22% | -4.99*** | -9.30% | -3.21*** |
| Home Furnishing Retail | -2.36% | -0.95 | 0.75% | 0.30 | 1.68% | 0.68 | -4.06% | -1.64 | -1.06% | -0.43 | 2.43% | 0.69 | -2.69% | -0.54 |
| Textiles, Apparel and Luxury Goods | -1.16% | -0.72 | 1.08% | 0.67 | -0.69% | -0.43 | -1.29% | -0.80 | -0.41% | -0.25 | 0.39% | 0.17 | -1.31% | -0.40 |
| **Consumer Staples** | 0.43% | 0.84 | -1.22% | -2.39** | 2.02% | 3.96*** | 0.35% | 0.69 | 0.39% | 0.76 | 0.80% | 1.11 | 1.54% | 1.51 |
| Food and Staples Retailing | 0.47% | 0.56 | 0.72% | 0.86 | -2.29% | -2.73*** | 1.47% | 1.75 | 0.28% | 0.33 | -0.82% | -0.69 | 0.67% | 0.80 |
| Drug Retail | -0.25% | -0.14 | 5.90% | 3.19*** | 6.67% | 3.61*** | -2.41% | -1.30 | -0.59% | -0.32 | 12.57% | 4.80*** | 9.57% | 2.59*** |
| Food Retail | 1.39% | 0.97 | -7.05% | -4.93*** | 2.95% | 2.06** | 0.32% | 0.22 | 0.11% | 0.08 | -4.01% | -2.03** | -3.67% | -1.28 |
| Hypermarkets and Super Centers | 0.78% | 0.75 | -4.11% | -3.95*** | 0.89% | 0.86 | 1.11% | 1.07 | 0.82% | 0.79 | -3.22% | -2.19** | -1.29% | -0.62 |
| Brewers | 0.38% | 0.22 | 6.62% | 3.78*** | 4.17% | 2.38** | -1.30% | -0.74 | 0.39% | 0.22 | 10.79% | 4.36*** | 9.88% | 2.82*** |
| Soft Drinks | -1.04% | -1.42 | 0.11% | 0.15 | 1.86% | 2.56*** | 2.45% | 2.56*** | 0.01% | 0.01 | 4.31% | 4.17*** | 0.10% | 0.14 |
| Food Products | 0.45% | 0.66 | -1.65% | -2.43** | 1.77% | 2.60*** | 0.77% | 1.13 | 0.19% | 0.28 | 0.12% | 0.12 | 1.08% | 0.79 |
| Agricultural Products | -0.33% | -0.34 | 0.04% | 0.04 | 2.14% | 2.18** | -0.78% | -0.80 | -0.65% | -0.66 | 2.18% | 1.57 | 0.75% | 0.38 |
| Packaged Foods and Meats | 0.44% | 0.64 | -1.92% | -2.78*** | 1.79% | 2.59*** | 0.93% | 1.35 | 0.31% | 0.45 | -0.13% | -0.13 | 1.11% | 0.80 |
| Household Products | 0.60% | 0.95 | -4.54% | -7.21*** | 1.78% | 2.83*** | 0.88% | 1.40 | 0.61% | 0.97 | -2.76% | -3.09*** | -1.27% | -1.00 |
| Personal Products | 0.68% | 0.52 | 2.12% | 1.62 | 0.48% | 0.37 | -0.33% | -0.25 | 1.20% | 0.92 | 2.60% | 1.40 | 3.47% | 1.32 |
| Tobacco | -0.34% | -0.27 | 1.41% | 1.10 | 4.18% | 3.27*** | 0.14% | 0.11 | -0.15% | -0.12 | 5.59% | 3.09*** | 5.58% | 2.72*** |
| **Industrials** | 0.11% | 0.13 | 2.20% | 2.50** | 1.85% | 2.10** | -1.61% | -1.83 | -0.32% | -0.36 | 4.05% | 3.25*** | 2.12% | 1.20 |
| Airlines | 4.46% | 1.44 | 1.79% | 0.58 | 10.06% | 3.25*** | 0.01% | 0.00 | -4.46% | -1.44 | 11.85% | 2.70*** | 20.36% | 1.98** |
| Railroads | -1.02% | -0.89 | 2.24% | 1.96** | 2.36% | 2.05** | -1.61% | -1.40 | 0.53% | 0.46 | 4.60% | 2.83*** | 3.52% | 1.53 |
| Transportation Infrastructure | -0.98% | -0.84 | 5.93% | 5.07*** | 0.55% | 0.47 | -2.27% | -1.94 | -1.09% | -0.93 | 6.48% | 3.91*** | 3.12% | 1.33 |
| Air Freight and Logistics | 1.27% | 0.82 | -4.75% | -3.08*** | 0.71% | 0.46 | -0.46% | -0.30 | -0.94% | -0.61 | -4.04% | -1.86* | -5.44% | -1.77* |
| Building products | 0.87% | 0.84 | -1.97% | -1.91* | -0.70% | -0.68 | -0.78% | -0.76 | 0.27% | 0.26 | -2.67% | -1.83* | -3.18% | -1.54 |
| Aerospace and Defense | -0.07% | -0.05 | 5.92% | 4.00*** | 3.71% | 2.51** | -2.61% | -1.76 | -0.46% | -0.31 | 9.63% | 4.60*** | 6.56% | 2.22** |
| Electrical Equipment | -0.01% | -0.01 | 3.49% | 3.17*** | 0.81% | 0.74 | -2.49% | -2.26** | -0.45% | -0.41 | 4.30% | 2.76*** | 1.36% | 0.62 |
| Industrial Conglomerates | 0.54% | 0.45 | 3.30% | 2.73*** | 3.00% | 2.48** | -1.41% | -1.17 | -0.48% | -0.40 | 6.30% | 3.68*** | 4.41% | 1.82* |
| Machinery | -0.12% | -0.11 | 1.55% | 1.48 | 1.50% | 1.43 | -1.77% | -1.69 | -0.22% | -0.21 | 3.05% | 2.05** | 1.06% | 0.51 |
| **Information Technology** | 0.32% | 0.44 | -2.27% | -3.11*** | -1.78% | -2.44** | 1.37% | 1.88 | 0.37% | 0.51 | -4.05% | -3.92*** | -2.31% | -1.58 |
| Communications Equipment | 1.40% | 1.05 | 1.52% | 1.14 | 1.06% | 0.80 | 1.29% | 0.97 | -0.48% | -0.36 | 2.58% | 1.37 | 3.39% | 1.27 |
| IT Services | 0.06% | 0.10 | 2.35% | 3.79*** | -0.11% | -0.18 | -0.19% | -0.31 | -0.55% | -0.89 | 2.24% | 2.55** | 1.50% | 1.21 |
| IT Consulting and Other Services | 0.21% | 0.24 | 0.62% | 0.71 | 0.89% | 1.02 | 0.20% | 0.23 | -0.95% | -1.09 | 1.51% | 1.23 | 0.76% | 0.44 |
| Data Processing and Outsourcing | 0.05% | 0.07 | 2.97% | 4.13*** | -0.33% | -0.46 | -0.31% | -0.43 | -0.48% | -0.67 | 2.64% | 2.59*** | 1.85% | 1.28 |
| Internet Services and Infrastructure | -0.75% | -0.60 | -2.86% | -2.29*** | -1.87% | -1.50 | 0.19% | 0.15 | 0.71% | 0.57 | -4.73% | -2.67*** | -3.83% | -1.53 |

(*Continued*)

**Table 2.** (Continued)

| Event Windows | [-1] | | [0] | | [1] | | [2] | | [3] | | [0, 1] | | [0, 3] | |
|---|---|---|---|---|---|---|---|---|---|---|---|---|---|---|
| Indices | AR | T-Stats. | AR | T-Stats. | AR | T-Stats. | AR | T-Stats. | AR | T-Stats. | CAR | T-Stats. | CAR | T-Stats. |
| Semiconductors and Equipment | 1.36% | 1.33 | -3.38% | -3.31*** | -3.36% | -3.29*** | 2.63% | 2.58*** | 0.13% | 0.13 | -6.74% | -4.67*** | -3.98% | -1.95 |
| Software | 0.27% | 0.24 | -3.78% | -3.35*** | -3.44% | -3.04*** | 1.61% | 1.42 | 0.68% | 0.60 | -7.22% | -4.52*** | -4.93% | -2.18** |
| Technology Hardware and Storage | -0.35% | -0.20 | -3.81% | -2.15** | -0.18% | -0.10 | 1.44% | 0.81 | 1.14% | 0.64 | -3.99% | -1.59 | -1.41% | -0.40 |
| **Materials** | 0.12% | 0.14 | 1.10% | 1.25 | 1.18% | 1.34 | -2.14% | -2.43** | -1.35% | -1.53 | 2.28% | 1.83* | -1.21% | -0.69 |
| Chemicals | 0.23% | 0.24 | 1.88% | 1.94* | 1.38% | 1.42 | -2.72% | -2.80*** | -1.50% | -1.55 | 3.26% | 2.38** | -0.96% | -0.49 |
| Construction Materials | -1.97% | -1.13 | 0.25% | 0.14 | 2.42% | 1.39 | -0.18% | -0.10 | -1.32% | -0.76 | 2.67% | 1.09 | 1.17% | 0.34 |
| Containers and Packaging | -0.10% | -0.10 | -0.41% | -0.41 | 1.30% | 1.29 | -0.81% | -0.80 | -1.88% | -1.86 | 0.89% | 0.62 | -1.80% | -0.89 |
| Metals and Mining | 0.50% | 0.34 | -1.33% | -0.92 | -0.52% | -0.36 | -1.01% | -0.70 | 0.03% | 0.02 | -1.85% | -0.90 | -2.83% | -0.98 |
| **Utilities** | -0.30% | -0.31 | 1.17% | 1.21 | 1.40% | 1.44 | -0.10% | -0.10 | -1.35% | -1.39 | 2.57% | 1.87* | 1.12% | 0.58 |
| Electric Utilities | -0.20% | -0.20 | 0.88% | 0.88 | 1.11% | 1.11 | 0.31% | 0.31 | -1.26% | -1.26 | 1.99% | 1.41 | 1.04% | 0.52 |
| Gas Utilities | -1.36% | -1.26 | 4.08% | 3.78*** | 4.79% | 4.44*** | -1.82% | -1.69 | -1.61% | -1.49 | 8.87% | 5.81*** | 5.44% | 2.52** |
| Water Utilities | 1.66% | 1.44 | -0.01% | -0.01 | 0.61% | 0.53 | -0.03% | -0.03 | -1.52% | -1.32 | 0.60% | 0.37 | -0.95% | 0.41 |
| Multi Utilities | -0.53% | -0.49 | 1.75% | 1.62 | 2.00% | 1.85* | -0.92% | -0.85 | -1.38% | -1.28 | 3.75% | 2.45** | 1.45% | 0.67 |

Note: This table shows the Market Model Abnormal Returns and Cumulative Abnormal Returns for different event windows. The null hypothesis of H0: AR = 0 orCAR = 0 is tested with t-test.

***,**,* indicate 1%, 5%, and 10% significance,respectively. Data is obtained from Thomson Reuters Eikon for the period 23 June 2020 to 12 November 2020, for a total of 139 daily observations per index.

Financial industry benefits from positive economic news and a healthy economic environment that the availability of a vaccine is expected to promote. Some indicators such as the US 10-year treasury bond yields increased to 1.675% after the vaccine announcement which is the highest level since March 2020. The vaccine understandably provides an opportunity for the economy to begin recovering.

The energy sector gains 12.45 percent and 12.34 percent on [0] and [0, 3] respectively. The four sub-sectors all show gains of over 13 percent on [0]. The S&P energy sector is by far the worst performing index of the eleven sectors during the pandemic [4]. It is unsurprising that the energy sector and oil and gas subsectors show a positive response. Demand for energy would be expected to increase with the easing of restrictions and return to economic normalcy enabled by the vaccine rollout. In fact IMF projected a global growth of 6.0 percent in 2021 due to vaccine rollout and stimuli [34]. Though the real estate management response is muted, the real estate management and development subsector is more upbeat about the vaccine with a gain of 9.32 percent on [0], and 6.60 percent over [0, 1]. The subsector has been hit hard by a global work-from-home trend. A vaccine raises the prospect of people returning to the office. The real estate development would also benefit from a growing economy. The communications services sector experiences a moderate loss of 1.54 percent. Subsectors that offer news, broadcasting and entertainment services indicate loss, possibly because it is expected that home consumption of these services would decline once economic activity fully resume. However, more integrated/diversified telecommunication services that are linked to the overall economic performance, show some gain.

Overall, the health care sector does not show significant reaction to the announcement, but some subsectors such as biotechnology and life sciences tools, engaged mainly in the research, development, and manufacturing, register losses. This likely reflects the lost competitive edge

of firms that are in competition with the developers of Pfizer. In addition, health care providers such as distributors and operators of health care facilities indicate a positive response to the announcement. COVID-19 pandemic has significantly disrupted services for noncommunicable diseases [35]. A return to economic normalcy would allow health care providers to meet this pent-up demand. Furthermore, health care providers would benefit if involved in the vaccine rollout.

The consumer discretionary sector records a negative impact of 2.91 percent and 3.56 percent at [0] and [0, 3] respectively. But the subsector reactions vary significantly. Hotels register a substantial gain of 23.97 percent over [0, 3] window. [4] show that entertainment and hospitality were negatively affected by the COVID-19 pandemic. These subsectors would benefit from elimination of restrictions. Home building, household appliances and computers and electronics retail subsectors lost 9.21 percent, 11.35 percent and 9.3 percent over [0, 3] window, respectively as vaccination would reverse the work-from-home trend. The announcement triggered significant disparate effects within the subsectors of consumer staples. On the event day, food retail and hypermarket sub-sectors lost. So do packaged foods, and household products. This can be attributed to several reasons. One of the main drivers is the anticipated reduced work-from-home when the COVID-19 pandemic is controlled by widespread vaccination. Also, with the economic activity expected to go back to normal, consumer uncertainty, fear and thus panic buying would decrease. Literature shows that perceived scarcity drives panic buying [36, 37]. Furthermore studies [38, 39] relate panic buying to social media use. In fact, some literature [40] have found that social influence can produce a stronger impact on panic buying than perceived scarcity. With the announcement of the vaccine, social media likely would have more upbeat messages. Leisure good providers like brewers gained 9.88 percent over [0, 3].

The airline industry which stands to benefit from resumption of travel show substantial gains of 20.36 percent over [0, 3]. [41] show that airline industry stocks were affected more than the market average by the pandemic. Railroad and transportation infrastructure also show (more moderate) gains. Increase in e-commerce spurred by COVID-19 restrictions, and the requirement for essential medical supplies by governments to meet the challenges of COVID-19 stimulated air freight demand. This explains the 5.44 percent loss over [0, 3] by air freight business on vaccine announcement.

Return to normal work environment would limit demand for internet, software and other products needed to operate from home or remote work locations. These subsectors show negative reaction to the announcement. However, IT services such as consulting and information management, and system integration (IT services), as well as data processing rely more on the performance of the overall economy. This explains why they show a positive reaction to the announcement. The effect on materials sector is muted with chemicals subsector having a modest gain of 1.88 percent on [0]. Utilities sector also does not display significant reaction except for the gas utilities subsector that gains 5.44 percent over [0, 3]. Gas utilities comprise primarily companies that distribute natural and manufactured gas. Low prices that encouraged switching from coal to gas for power generation, and strong demand from China and Asia-Pacific region in 2020 are the main reasons that affect gas stocks positively. Anticipated improvement in the general economy due to a vaccine rollout would also positively affect demand for gas.

## 5.1 Robustness analysis

As the period chosen from June to October 2020 represents the recovery phase after COVID-19 crisis, we choose an additional pre-pandemic estimation period between 10 September

2019 to 16 January 2020 for a robustness analysis following [42, 43]. The results obtained (see Table 3) closely mirror the results in Table 2 indicating the results are robust to the choice of estimation period. The financial sector gains with consumer finance and banks gaining the most. Likewise the gains by energy sector are also large. The real estate management and development subsector shows a gain of 9.06 percent over [0], and 6.46 percent over [0, 1]. Overall, the communications services sector lost 1.71 percent. As in Table 2, subsectors that offer news, broadcasting and entertainment services indicate loss, while more integrated/diversified telecommunication services that are linked to the overall economic performance, show some gain. Reactions in consumer discretionary show strong gains by hotels and casinos, but losses by direct retail, internet, computers and electronics. As in Table 2 food retail and hypermarket show losses, while brewers and drug retail gained. Airline industry and transportation show substantial gains of 12.54 and 5.93 percent on [0], while airfreight register losses. All in all, the results in Table 3 closely mimic the results in Table 2.

As a further robustness check, a battery of t-tests are carried out to test if the magnitude of the market reaction (ARs and CARs) is different across the 11 sectors and 79 subsectors. First, in Tables 4 to 14, the t-tests are carried out at subsector level for each industry over [0] and [1]. The average effect for each subsector is averaged over [0], and [1] and compared. Overall, substantial differences are detected in subsector AR responses. Table 4 for example shows that within the finance sector, over [0] and [1], the average AR for the capital markets is significantly different from the insurance subsector and diversified finance service subsector. In the communication services (Table 7), the magnitude of AR on wireless telecommunication services is significantly lower than integrated telecommunication services and diversified telecommunication services, while AR for interactive media is smaller than for broadcasting. In consumer discretionary industry (Table 9), hotel subsector is one of the subsectors showing a relatively high positive response. It is found that the AR for hotel subsector over [0] and [1] is significantly different (higher) from some of the other subsectors such as the internet and direct retail and multiline retail. Auto components and automobiles are also significantly different form several sub sectors. Brewers in consumer staples (Table 10) is different from food and staples. Several subsectors also show differences in information technology (Table 12). The tests shows that a considerable number of subsector ARs display significant differences.

Finally, Table 15 shows tests of equality of mean abnormal returns averaged over the subsectors by industry on [0], while Table 16 follows the same approach to test for equality of average mean CAR (averaged over subsectors) by industry for [0, 3] interval. The results in Table 15 indicate the mean ARs for several industries differ (see the P-values reflected in the tables). In particular, the mean AR for the subsectors within the financial sector included in the analysis is significantly different from the means of several other sectors such as energy, consumer services, health care, information technology, and materials. Similarly, the average performance of energy sector which exhibited one of the highest positive reactions is different from nearly all the other sectors. Table 16 which tests for equality of the CARs for [0, 3] shows a similar trend. Financials, energy and to a lesser degree the real estate subsectors have CARs that significantly differ from other industries. Financials for example differ from energy, communication services, information technology and materials. While this supports the earlier results that there are differences in sector reactions to the vaccine announcement, the sector mean averages out some subsector differences, and might hide important differences at this level. Hence it is important to consider this part of the results with that obtained in Tables 4 to 14 at the subsector level above. Overall, the t-tests clearly show that there are significant differences in ARs and CARs at subsector and industry levels. This conforms with the results contained in Tables 2 and 3.

**Table 3. Market model abnormal return and cumulative abnormal return for different event windows-estimation period: 10 September 2019–16 January 2020.**

| Event Windows | [-1] | | [0] | | [1] | | [2] | | [3] | | [0, 1] | | [0, 2] | |
|---|---|---|---|---|---|---|---|---|---|---|---|---|---|---|
| Indices | AR | T-Stats. | AR | T-Stats. | AR | T-Stats. | AR | T-Stats. | AR | T-Stats. | CAR | T-Stats. | CAR | T-Stats. |
| **Financials** | -0.73% | -0.70 | 6.50% | 15.85*** | 0.89% | 2.17** | -1.39% | -3.39*** | -0.53% | -1.29 | 7.39% | 12.75*** | 5.47% | 6.67*** |
| Banks | -1.80% | -2.47** | 10.91% | 14.94*** | -0.01% | -0.01 | -1.97% | -2.70*** | -0.88% | -1.21 | 10.90% | 10.56*** | 8.05% | 5.51*** |
| Insurance | 0.17% | 0.38 | 4.40% | 9.78*** | 2.36% | 5.24*** | -1.62% | -3.60*** | -0.25% | -0.56 | 6.76% | 10.62*** | 4.89% | 5.43*** |
| Capital Markets | -0.80% | -2.16** | 1.74% | 4.70*** | 0.04% | 0.11 | 0.07% | 0.19 | -0.69% | -1.86* | 1.78% | 3.40*** | 1.16% | 1.57 |
| Consumer Finance | -1.32% | -2.28** | 13.64 | 23.52 | -0.49 | -0.84 | -4.13 | -7.12 | 0.76 | 1.31 | 13.15 | 16.03 | 9.78 | 8.43 |
| Diversified Financial Services | 0.69% | 1.50 | 4.74% | 10.30*** | 3.22% | 7.00*** | -1.24% | -2.70*** | -0.22% | -0.48 | 7.96% | 12.24*** | 6.50% | 7.07*** |
| **Energy** | -1.98% | -2.83*** | 12.86% | 18.37*** | 2.73% | 3.90*** | -1.08% | -1.54 | -2.76% | -3.94*** | 15.59% | 15.75*** | 11.75% | 8.39*** |
| Oil& Gas Exploration and Production | -3.03% | -1.99** | 16.52% | 10.87*** | 2.86% | 1.88* | 0.21% | 0.14 | -5.22% | -3.34*** | 19.38% | 9.02*** | 14.37% | 4.73*** |
| Oil & Gas Equipment Services | -1.97% | -1.17 | 18.13% | 10.79*** | 0.79% | 0.47 | -0.78% | -0.46 | -3.44% | -2.05** | 18.92% | 7.96*** | 14.70% | 4.38*** |
| Oil & Gas Drilling | -2.61% | -1.13 | 16.83 | 7.25 | -0.04 | -0.02 | -0.65 | -0.28 | -5.53 | -2.38 | 16.79 | 5.12 | 10.61 | 2.29 |
| Energy Equipment and Services | -1.73% | -1.40 | 18.19% | 14.66*** | 0.02% | 0.02 | -1.44% | -1.16 | -2.37% | -1.91* | 18.21% | 10.38*** | 14.40% | 5.81*** |
| **Real Estate** | -0.36% | -0.46 | 2.10% | 2.69*** | 0.68% | 0.87 | 0.66% | 0.85 | -0.72% | -0.92 | 2.78% | 2.52** | 2.72% | 1.74* |
| Equity Real Estate Investment Trusts | -0.36% | -0.46 | 2.10% | 2.69*** | 0.68% | 0.87 | 0.66% | 0.85 | -0.72% | -0.92 | 2.78% | 2.52** | 2.72% | 1.74* |
| Real Estate Mng.and Dev. | -1.17% | -1.14 | 9.06% | 8.80*** | -2.60% | -2.52** | -1.43% | -1.39 | -0.42% | -0.41 | 6.46% | 4.43*** | 4.61% | 2.24** |
| **Communication Services** | -0.12% | -0.35 | -1.71% | -5.02*** | -0.22% | -0.65 | -0.41% | -1.21 | 0.66% | 1.94* | -1.93% | -4.01*** | -1.68% | -2.47** |
| Wireless Telecommunication Services | 5.02% | 9.30*** | -1.22% | -2.25** | -1.05% | -1.94* | 2.32% | 4.29*** | 1.42% | 2.63*** | -2.27% | -2.97*** | 1.47% | 1.36 |
| Interactive Media and Services | -0.45% | -0.71 | -4.28 | -6.79*** | -1.69 | -2.68 | -0.63 | -1.00 | 1.22 | 1.94 | -5.97 | -6.70 | -5.38 | -4.27 |
| Broadcasting | -3.96% | -5.42*** | 1.04% | 1.42 | 3.39% | 4.64*** | 0.11% | 0.15 | -3.25% | -4.45*** | 4.43% | 4.29*** | 1.28% | 0.88 |
| Interactive Home Entertainment | -2.55% | -3.40*** | -4.84% | -6.45*** | -0.23% | -0.31 | 1.62% | 2.16** | 0.10% | 0.13 | -5.07% | -4.78*** | -3.35% | -2.23** |
| Media and Entertainment | -0.36% | -0.90 | -2.61% | -6.53*** | -0.63% | -1.58 | -0.63% | -1.58 | 0.90% | 2.25** | -3.24% | -5.73*** | -2.97% | -3.71*** |
| Integrated Telecommunication Services | 0.46% | 0.88 | 2.98% | 5.73*** | 2.20% | 4.23*** | 0.05% | 0.10 | -0.57% | -1.10 | 5.18% | 7.04*** | 4.66% | 4.48*** |
| Diversified Telecommunication Services | 0.40% | 0.83 | 2.92% | 6.08*** | 2.28% | 4.75*** | -0.0001% | -0.0004 | -0.54% | -1.13 | 5.20% | 7.66*** | 4.66% | 4.85*** |
| **Healthcare** | 0.14% | 0.31 | 0.07% | 0.16 | 0.40% | 0.89 | -0.60% | -1.33 | 0.12% | 0.27 | 0.47% | 0.74 | -0.01% | -0.01 |
| Biotechnology | -0.44% | -0.75 | -2.35% | -3.98*** | 2.05% | 3.47*** | -0.40% | -0.68 | 0.06% | 0.10 | -0.30% | -0.36 | -0.64% | -0.54 |
| Health Care Equipment and Supplies | 0.75% | 2.14** | 0.73% | 2.09** | -0.79% | -2.26** | -0.59% | -1.69* | 0.25% | 0.71 | -0.06% | -0.12 | -0.40% | -0.57 |
| Health Care Distributors | -0.94% | -0.68 | 3.87% | 2.78*** | 2.32% | 1.67* | -3.97% | -2.86*** | -0.44% | -0.32 | 6.19% | 3.15*** | 1.78% | 0.64 |
| Heath Care Facility | 0.32% | 0.50 | 6.85% | 10.70*** | 3.40% | 5.31*** | -3.84% | -6.00*** | -1.12% | -1.75 | 10.25% | 11.32*** | 5.29% | 4.13*** |
| Health Care Technology | 0.66% | 0.94 | 0.95% | 1.36 | 1.05% | 1.50 | -0.38% | -0.54 | -0.97% | -1.39 | 2.00% | 2.02** | 0.65% | 0.46 |
| Life Sciences Tools and Services | 2.07% | 3.83*** | -5.71% | -10.57*** | -3.08% | -5.71*** | 1.58% | 2.93*** | 1.32% | 2.44** | -8.79% | -11.51*** | -5.89% | -5.45*** |
| Pharmaceuticals | 0.20% | 0.48 | 1.38% | 3.28*** | 0.64% | 1.52 | -0.95% | -2.26** | -0.51% | -1.21 | 2.02% | 3.40*** | 0.56% | 0.67 |

(*Continued*)

**Table 3.** (Continued)

| Event Windows | [-1] | | [0] | | [1] | | [2] | | [3] | | [0, 1] | | [0, 2] | |
|---|---|---|---|---|---|---|---|---|---|---|---|---|---|---|
| Indices | AR | T-Stats. | AR | T-Stats. | AR | T-Stats. | AR | T-Stats. | AR | T-Stats. | CAR | T-Stats. | CAR | T-Stats. |
| **Consumer Discretionary** | -0.19% | -0.53 | -2.90% | -8.05*** | -0.91% | -2.53** | 0.66% | 1.83* | 0.02% | 0.05 | -3.81% | -7.48*** | -3.13% | -4.35*** |
| Auto Components | 0.15% | 0.13 | 1.69% | 1.50 | 2.94% | 2.60*** | 1.01% | 0.89 | -0.85% | -0.75 | 4.63% | 2.90*** | 4.79% | 2.12** |
| Automobiles | 0.07% | 0.10 | 2.66% | 3.86*** | 4.17% | 6.04*** | -0.82% | -1.18 | -0.62% | -0.90 | 6.83% | 6.99*** | 5.39% | 3.91*** |
| Hotels | 0.18% | 0.30 | 15.99% | 26.21*** | -1.73% | -2.84*** | -2.47% | -4.04*** | -1.03% | -1.69* | 14.26% | 16.53*** | 10.76% | 8.82*** |
| Casinos and Gaming | 0.56% | 0.80 | 12.10% | 17.29*** | -1.18% | -1.69* | -4.71% | -6.73*** | -1.42% | -2.03** | 10.92% | 11.03*** | 4.79% | 3.42*** |
| Restaurants | -0.01% | -0.02 | -0.20% | -0.43 | -1.01% | -2.20** | 0.59% | 1.28 | -1.09% | -2.37** | -1.21% | -1.86* | -1.71% | -1.86* |
| Household Durables | -2.42% | -3.84*** | -5.30% | -8.41*** | 3.75% | 5.95*** | 0.75% | 1.19 | -2.23% | -3.54*** | -1.55% | -1.74* | -3.03% | -2.40** |
| Home Building | -4.01% | -5.01*** | -6.97% | -8.71*** | 5.98% | 7.47*** | 1.79% | 2.24** | -3.37% | -4.21*** | -0.99% | -0.88 | -2.57% | -1.61 |
| Household Appliances | 1.25% | 2.02** | -12.72% | -20.51*** | 1.25% | 2.02** | -0.10% | -0.16 | 0.44% | 0.71 | -11.47% | -13.08*** | -11.13% | -8.98*** |
| Consumer Electronics | 1.91% | 4.15*** | -0.64% | -1.39 | -0.63% | -1.37 | 0.16% | 0.35 | 0.06% | 0.13 | -1.27% | -1.95* | -1.05% | -1.14 |
| Internet and Direct Marketing Retail | -0.09% | -0.08 | -6.05% | -5.35*** | -3.33% | -2.95*** | 1.67% | 1.48 | 0.94% | 0.83 | -9.38% | -5.87*** | -6.77% | -3.00*** |
| Multiline Retail | 1.78% | 1.66* | -3.06% | -2.86*** | 1.12% | 1.05 | 1.50% | 1.40 | -0.49% | -0.46 | -1.94% | -1.28 | -0.93% | -0.43 |
| Distributors | -0.79% | -1.05 | -4.05% | -5.40*** | 4.11% | 5.48*** | -0.99% | -1.32 | -0.96% | -1.28 | 0.06% | 0.06 | -1.89% | -1.26 |
| General Merchandise Stores | 1.83% | 1.54 | -3.12% | 2.62*** | 1.18% | 0.99 | 1.47% | 1.24 | -0.35% | -0.29 | -1.94% | -1.15 | -0.82% | -0.34 |
| Specialty Retail | -0.45% | -1.25 | -2.95% | -8.19*** | 1.88% | 5.22*** | -0.47% | -1.31 | 0.11% | 0.31 | -1.07% | -2.10** | -1.43% | -1.99** |
| Computers and Electronics Retail | -0.09% | -0.10 | -10.33% | -11.61*** | 1.36% | 1.53 | 1.66% | 1.87* | -0.64% | -0.71 | -8.97% | -7.13*** | -7.95% | -4.47*** |
| Home Furnishing Retail | -2.38% | -1.90* | 1.07% | 0.86 | 1.63% | 1.30 | -3.85% | -3.08*** | -1.35% | -1.08 | 2.70% | 1.53 | -2.50% | -1.00 |
| Textiles, Apparel and Luxury Goods | -0.97% | -1.59 | 1.47% | 2.41** | -0.53% | -0.87 | -0.97% | -1.59 | -0.39% | -0.64 | 0.94% | 1.09 | -0.42% | -0.34 |
| **Consumer Staples** | 0.45% | 0.94 | -0.96% | -2.00** | 2.02% | 4.21*** | 0.53% | 1.10 | 0.21% | 0.44 | 1.06% | 1.56 | 1.80% | 1.88* |
| Food and Staples Retailing | 0.98% | 2.72*** | -1.64% | -4.56*** | 1.69% | 4.69*** | 0.79% | 2.19** | 0.61% | 1.69* | 0.05% | 0.10 | 1.45% | 2.01** |
| Drug Retail | -0.22% | -0.12 | 7.67% | 4.24*** | 6.54% | 3.61*** | -1.22% | -0.67 | -1.98% | -1.09 | 14.21% | 5.55*** | 11.01% | 3.04*** |
| Food Retail | 1.46% | 1.74* | -5.89% | -7.01*** | 2.91% | 3.46*** | 1.12% | 1.33 | -0.71% | -0.85 | -2.99% | -2.51*** | -2.57% | -1.53 |
| Hypermarkets and Super Centers | 1.07% | 2.33** | -3.77% | -9.20*** | 1.19% | 2.59*** | 1.44% | 3.13*** | 1.08% | 2.35** | -2.58% | -3.97*** | -0.06% | -0.06 |
| Brewers | -0.11% | -0.14 | 6.52% | 8.58*** | 3.64% | 4.79*** | -1.53% | -2.01** | -0.43% | -0.57 | 10.16% | 9.45*** | 8.20% | 5.39*** |
| Soft Drinks | | | 1.90% | 2.97*** | 2.32% | 3.63*** | 0.01% | 0.02 | -0.14% | -0.22 | 4.22% | 4.66*** | 4.09% | 3.20*** |
| Food Products | 0.40% | 0.57 | -0.62% | -0.89 | 1.63% | 2.33** | 1.44% | 2.06** | -0.74% | -1.06 | 1.01% | 1.02 | 1.71% | 1.22 |
| Agricultural Products | -0.33% | -0.34 | -1.43% | -1.86* | 2.65% | 3.44*** | -2.98% | -3.87*** | 1.77% | 2.30** | 1.22% | 1.12 | 0.01% | 0.006 |
| Packaged Foods and Meats | 0.34% | 0.47 | -0.90% | -1.23 | 1.59% | 2.18** | 1.57% | 2.15** | -0.70% | -0.96 | 0.69% | 0.67 | 1.56% | 1.07 |
| Household Products | 0.68% | 1.00 | -4.52% | -6.65*** | 1.86% | 2.73*** | 0.91% | 1.34 | 0.73% | 1.07 | -2.66% | -2.76*** | -1.02% | -0.75 |
| Personal Products | 0.50% | 0.68 | 1.64% | 2.24** | 0.33% | 0.45 | -0.71% | -0.97 | 1.26% | 1.73* | 1.97% | 1.91* | 2.52% | 1.73* |
| Tobacco | -0.49% | -0.77 | 1.19% | 1.86* | 4.03% | 6.29*** | -0.06% | -0.09 | -0.25% | -0.39 | 5.22% | 5.77*** | 4.91% | 3.84*** |
| **Industrials** | 0.15% | 0.38 | 2.15% | 5.38*** | 1.90% | 4.75*** | -1.63% | -4.08*** | -0.21% | -0.53 | 4.05% | 7.16*** | 2.21% | 2.76*** |
| Airlines | -1.89% | -2.20** | 12.54% | 14.58*** | 0.15% | 0.17 | -4.87% | -5.66*** | -1.63% | -1.90* | 12.69% | 10.43*** | 6.19% | 3.60*** |
| Railroads | -1.02% | -0.89 | 2.16% | 4.59*** | 2.42% | 5.15*** | -1.65% | -3.51*** | 0.68% | 1.45 | 4.58% | 6.89*** | 3.61% | 3.84*** |
| Transportation Infrastructure | -0.99% | -1.90* | 5.93% | 11.40*** | 0.54% | 1.04 | -2.27% | -4.37*** | -1.11% | -2.13** | 6.47% | 8.80*** | 3.09% | 2.97*** |
| Air Freight and Logistics | 1.67% | 2.88*** | -4.32% | -7.45*** | 1.12% | 1.93* | -0.04% | -0.07 | -0.56% | -0.97 | -3.20% | -3.90*** | -3.80% | -3.28*** |
| Building products | 1.25% | 4.17*** | -2.00% | -6.66*** | -0.29% | -0.97 | -0.67% | -2.23** | 0.98% | 3.27*** | -2.29% | -5.40*** | -1.98% | -3.30*** |

*(Continued)*

**Table 3.** (Continued)

| Event Windows | [-1] | | [0] | | [1] | | [2] | | [3] | | [0, 1] | | [0, 2] | |
|---|---|---|---|---|---|---|---|---|---|---|---|---|---|---|
| Indices | AR | T-Stats. | AR | T-Stats. | AR | T-Stats. | AR | T-Stats. | AR | T-Stats. | CAR | T-Stats. | CAR | T-Stats. |
| Aerospace and Defense | -0.44% | -0.50 | 6.04% | 6.86*** | 3.29% | 3.74*** | -2.65% | -3.01*** | -1.23% | -1.40 | 9.33% | 7.50*** | 5.45% | 3.10*** |
| Electrical Equipment | 0.19% | 0.48 | 3.56% | 8.90*** | 1.03% | 2.58*** | -2.37% | -5.93*** | -0.14% | -0.35 | 4.59% | 8.11*** | 2.08% | 2.60*** |
| Industrial Conglomerates | 0.55% | 0.71 | 2.40% | 3.08*** | 3.09% | 3.96*** | -2.00% | -2.56*** | 0.26% | 0.33 | 5.49% | 4.98*** | 3.75% | 2.40** |
| Machinery | 0.27% | 0.84 | 1.53% | 4.78*** | 1.93% | 6.03*** | -1.65% | -5.16*** | 0.50% | 1.56 | 3.46% | 7.65*** | 2.31% | 3.61*** |
| **Information Technology** | 0.27% | 1.08 | -2.64% | -10.56*** | -1.80% | -7.20*** | 1.11% | 4.44*** | 0.57% | 2.28** | -4.44% | -12.56*** | -2.76% | -5.52*** |
| Communications Equipment | 1.27% | 2.65*** | 0.83% | 1.73* | 0.97% | 2.02** | 0.79% | 1.65* | -0.17% | -0.35 | 1.80% | 2.65*** | 2.42% | 2.52** |
| IT Services | -0.10% | -0.28 | 1.85% | 5.14*** | -0.25% | -0.69 | -0.58% | -1.61 | -0.44% | -1.22 | 1.60% | 3.14*** | 0.58% | 0.81 |
| IT Consulting and Other Services | 0.34% | 0.85 | 0.67% | 1.68* | 1.03% | 2.58*** | 0.27% | 0.68 | -0.76% | -1.90* | 1.70% | 3.00*** | 1.21% | 1.51 |
| Data Processing and Outsourcing | -0.18% | -0.46 | 2.32% | 5.95*** | -0.52% | -1.33 | -0.82% | -2.10** | -0.37% | -0.95 | 1.80% | 3.26*** | 0.61% | 0.78 |
| Internet Services and Infrastructure | -1.44% | -1.87* | -3.63% | -4.71*** | -2.54% | -3.30*** | -0.55% | -0.71 | 0.09% | 0.12 | -6.17% | -5.66*** | -6.63% | -4.30*** |
| Semiconductors and Equipment | 1.60% | 2.02** | -3.17% | -4.01*** | -3.12% | -3.94*** | 2.85% | 3.61*** | 0.38% | 0.48 | -6.29% | -5.63*** | -3.06% | -1.94* |
| Software | 0.20% | 0.74 | -4.35% | -16.11*** | -3.46% | -12.81*** | 1.21% | 4.48*** | 1.03% | 3.81*** | -7.81% | -20.45*** | -5.57% | -10.31*** |
| Technology Hardware and Storage | -0.52% | -0.74 | -4.40% | -6.29*** | -0.31% | -0.44 | 0.99% | 1.41 | 2.31% | 1.87* | -4.71% | -4.76*** | -2.41% | -1.72* |
| **Materials** | 0.38% | 0.53 | 1.47% | 2.04** | 1.44% | 2.00** | -1.81% | -2.51** | -1.18% | -1.64* | 2.91% | 2.86*** | -0.08% | -0.06 |
| Chemicals | 0.49% | 0.60 | 2.09% | 2.58*** | 1.65% | 2.04** | -2.49% | -3.07*** | -1.19% | -1.47 | 3.74% | 3.26*** | 0.06% | 0.04 |
| Construction Materials | -1.50% | -1.40 | 1.08% | 1.01 | 2.86% | 2.67*** | 0.53% | 0.50 | -1.14% | -1.07 | 3.94% | 2.60*** | 3.33% | 1.56 |
| Containers and Packaging | 0.20% | 0.24 | 0.01% | 0.01 | 1.59% | 1.87* | -0.42% | -0.49 | -1.66% | -1.95* | 1.60% | 1.33 | -0.48% | -0.28 |
| Metals and Mining | 0.60% | 0.64 | -0.23% | -0.24 | -0.53% | -0.56 | -0.25% | -0.27 | -0.07% | -0.75 | -0.76% | -0.57 | -1.71% | -0.91 |
| **Utilities** | -0.44% | -0.71 | 1.70% | 2.74*** | 1.19% | 1.92 | 0.02% | 0.32 | -2.04% | -3.29*** | 2.89% | 3.30*** | 1.05% | 0.85 |
| Electric Utilities | -0.37% | -0.57 | 1.37% | 2.11** | 0.88% | 1.35 | 0.57% | 0.88 | -1.97% | -3.03*** | 2.25% | 2.45** | 0.85% | 0.65 |
| Gas Utilities | -1.75% | -2.50** | 4.64% | 6.63*** | 4.31% | 6.18*** | -1.58% | -2.25** | -2.78% | -3.97*** | 8.95% | 9.04*** | 4.59% | 3.28*** |
| Water Utilities | 1.48% | 1.49 | 0.72% | 0.72 | 0.35% | 0.35 | 0.39% | 0.39 | -2.43% | -2.45** | 1.07% | 0.76 | -0.97% | -0.49 |
| Multi Utilities | -0.65% | -1.12 | 2.31% | 3.98*** | 1.82% | 3.14*** | -0.58% | -1.00 | -2.07% | -3.57*** | 4.13% | 5.04*** | 1.48% | 1.28 |

Note: This table shows the Market Model Abnormal Returns and Cumulative Abnormal Returns for different event windows. The null hypothesis of H0: AR = 0 orCAR = 0 is tested with t-test.

***,**,* indicate 1%, 5%, and 10% significance,respectively. Data is obtained from Thomson Reuters Eikon for the period 23 June 2020 to 12 November 2020, for a total of 139 daily observations per index.

## 6 Conclusion and policy recommendations

COVID-19 pandemic caused an unprecedented level of uncertainty and fear. The announcement of a successful phase 3 trial of Pfizer vaccine on November 9 marks a major milestone in the fight to control the pandemic. Previous literature on the impact of vaccine development [13–15] has focused on the aggregate market level, and not at the sectoral level. In this study, the effect of the Pfizer vaccine announcement on S&P 500 11 sectors and a total of 79 subsectors is analysed. The results follow an interesting pattern. The results clearly indicate the announcement generates optimism in a range of subsectors, while some other subsectors are

**Table 4. Financials industry.**

|  | Banks | Insurance | Capital Markets | Consumer Finance | Diversified Fin.Serv. |
|---|---|---|---|---|---|
| Banks |  | **0.70** | **0.50** | **0.65** | **0.78** |
| Insurance | **0.70** |  | 0.04 | **0.69** | **0.22** |
| Capital Market | **0.50** | 0.04 |  | **0.53** | 0.02 |
| Consumer Finance | **0.65** | **0.69** | **0.53** |  | **0.75** |
| Diversified Fin.Serv. | **0.78** | **0.22** | **0.02** | **0.75** |  |

The table reports the p values of t-test checking the equality of means of AR(0) and AR(1) values. The null hypothesis states that the difference in group means is zero. An alternate hypothesis states that the difference in group means is different from zero. Data is obtained from Thomson Reuters Eikon for the period 23 June 2020 to 12 November 2020, for a total of 139 daily observations per index.

**Table 5. Energy industry.**

|  | Oil& Gas Exloration and Production | Oil & Gas Equipment Services | Oil & Gas Drilling | Energy Equipment and Services |
|---|---|---|---|---|
| Oil& Gas Exloration and Production |  | **0.92** | **0.72** | **0.86** |
| Oil & Gas Equipment Services | **0.92** |  | **0.33** | **0.66** |
| Oil & Gas Drilling | **0.72** | **0.33** |  | **0.82** |
| Energy Equipment and Services | **0.86** | **0.66** | **0.82** |  |

The table reports the p values of t-test checking the equality of means of AR(0) and AR(1) values. The null hypothesis states that the difference in group means is zero. An alternate hypothesis states that the difference in group means is different from zero. Data is obtained from Thomson Reuters Eikon for the period 23 June 2020 to 12 November 2020, for a total of 139 daily observations per index.

**Table 6. Real estate industry.**

|  | Equity Real Estate Investment Trusts | Real Estate Mng.and Dev. |
|---|---|---|
| Equity Real Estate Investment Trusts |  | **0.77** |
| Real Estate Mng.and Dev. | **0.77** |  |

The table reports the p values of t-test checking the equality of means of AR(0) and AR(1) values. The null hypothesis states that the difference in group means is zero. An alternate hypothesis states that the difference in group means is different from zero. Data is obtained from Thomson Reuters Eikon for the period 23 June 2020 to 12 November 2020, for a total of 139 daily observations per index.

**Table 7. Communication services industry.**

|  | Wire. Tel. Serv. | Inter. Media | Broadcasting | Interactive Home Ent. | Media and Ent. | Integ. Telecomm. | Div. Telecom. |
|---|---|---|---|---|---|---|---|
| Wireless Tel. Serv. |  | **0.25** | **0.15** | **0.61** | **0.66** | 0.09 | 0.08 |
| Interactive Media | **0.25** |  | 0.02 | **0.95** | **0.12** | **0.14** | **0.13** |
| Broadcasting | **0.15** | 0.02 |  | **0.20** | 0.07 | **0.89** | **0.87** |
| Interactive Home Ent. | **0.61** | **0.95** | **0.20** |  | **0.60** | **0.32** | **0.31** |
| Media and Entertainment | **0.66** | **0.12** | 0.07 | **0.60** |  | **0.15** | **0.13** |
| Integrated Telecomm. | 0.09 | **0.14** | **0.89** | **0.32** | **0.15** |  | **0.73** |
| Diversified Telecom. | 0.08 | **0.13** | **0.87** | **0.31** | **0.13** | **0.73** |  |

The table reports the p values of t-test checking the equality of means of AR(0) and AR(1) values. The null hypothesis states that the difference in group means is zero. An alternate hypothesis states that the difference in group means is different from zero. Data is obtained from Thomson Reuters Eikon for the period 23 June 2020 to 12 November 2020, for a total of 139 daily observations per index.

**Table 8. Health care industry.**

| | Biotechnology | Health Care Equipment and Supplies | Health Care Distributors | Heath Care Facility | Health Care Technology | Life Sciences Tools and Services | Pharmaceuticals |
|---|---|---|---|---|---|---|---|
| Biotechnology | | **0.77** | **0.42** | **0.40** | **0.69** | 0.09 | **0.72** |
| Health Care Equipment and Supplies | **0.77** | | 0.10 | **0.15** | **0.98** | **0.18** | **0.33** |
| Health Care Distributors | **0.42** | 0.10 | | **0.29** | 0.24 | **0.16** | 0.07 |
| Heath Care Facility | **0.40** | **0.15** | 0.29 | | 0.25 | 0.17 | **0.13** |
| Health Care Technology | **0.69** | **0.98** | 0.24 | 0.25 | | **0.11** | **0.80** |
| Life Sciences Tools and Services | 0.09 | **0.18** | **0.16** | **0.17** | 0.11 | | **0.19** |
| Pharmaceuticals | **0.72** | **0.33** | 0.07 | **0.13** | **0.80** | **0.19** | |

The table reports the p values of t-test checking the equality of means of AR(0) and AR(1) values. The null hypothesis states that the difference in group means is zero. An alternate hypothesis states that the difference in group means is different from zero. Data is obtained from Thomson Reuters Eikon for the period 23 June 2020 to 12 November 2020, for a total of 139 daily observations per index.

significantly depressed. The (sub)sectors that were hardest hit by the pandemic [4, 5] show the most gains from the vaccine news, while sectors that gained from the pandemic are depressed by the news. For example, financial sector, particularly consumer finance, energy, airlines, hotels, and casinos gain. Subsectors that gained from the COVID-19 environment and likely to lose from a return to normalcy lost. Examples include airfreight, home building, household

**Table 9. Consumer discretionary industry.**

| | 1 | 2 | 3 | 4 | 5 | 6 | 7 | 8 | 9 | 10 | 11 | 12 | 13 | 14 | 15 | 16 | 17 |
|---|---|---|---|---|---|---|---|---|---|---|---|---|---|---|---|---|---|
| Auto Comp. | | 0.01 | **0.12** | **0.67** | 0.32 | **0.66** | **0.78** | **0.45** | 0.09 | 0.06 | **0.37** | **0.65** | 0.37 | **0.46** | **0.46** | 0.08 | **0.46** |
| Automobiles | 0.01 | | **0.15** | **0.77** | 0.20 | **0.55** | **0.68** | **0.40** | 0.06 | 0.05 | **0.28** | **0.52** | 0.28 | **0.33** | **0.41** | 0.03 | **0.27** |
| Hotels | **0.12** | **0.15** | | **0.81** | 0.17 | 0.22 | 0.35 | 0.22 | 0.11 | 0.06 | **0.18** | 0.06 | 0.06 | **0.22** | 0.10 | **0.19** | |
| Casinos | **0.67** | **0.77** | **0.81** | | 0.51 | **0.67** | **0.72** | **0.56** | 0.52 | 0.42 | **0.57** | **0.67** | 0.57 | **0.62** | **0.57** | 0.65 | **0.53** |
| Restaurants | 0.32 | 0.20 | 0.17 | 0.51 | | 0.92 | 0.95 | **0.60** | 0.60 | 0.24 | **0.73** | 0.95 | 0.73 | 0.94 | 0.62 | 0.40 | 0.24 |
| Household Durables | **0.66** | 0.55 | 0.22 | **0.67** | 0.92 | | 0.97 | 0.22 | 0.97 | 0.80 | **0.73** | 0.80 | 0.89 | 0.21 | 0.71 | 0.87 | |
| Home Building | **0.78** | **0.68** | 0.35 | **0.72** | 0.95 | 0.97 | | 0.10 | 0.99 | 0.68 | **0.88** | 0.95 | 0.88 | 0.96 | 0.15 | 0.81 | **0.91** |
| Household App. | **0.45** | **0.40** | 0.22 | **0.56** | 0.60 | 0.22 | 0.10 | | 0.60 | 0.87 | **0.52** | 0.29 | 0.52 | 0.45 | 0.30 | 0.47 | **0.58** |
| Consumer Electr. | 0.09 | 0.06 | **0.11** | 0.52 | 0.60 | 0.97 | 0.99 | **0.60** | | 0.17 | **0.77** | 0.98 | 0.77 | 0.94 | 0.63 | 0.13 | **0.46** |
| Internet Ret. | 0.06 | 0.06 | 0.03 | 0.42 | 0.24 | 0.55 | 0.68 | 0.87 | 0.17 | | **0.36** | 0.45 | 0.36 | 0.23 | 0.93 | 0.06 | 0.24 |
| Multiline Retail | **0.37** | 0.28 | 0.06 | 0.57 | 0.73 | 0.80 | 0.88 | 0.52 | 0.77 | 0.36 | | 0.60 | 0.50 | 0.12 | 0.54 | 0.40 | 0.67 |
| Distributors | **0.65** | 0.52 | 0.18 | 0.67 | 0.95 | 0.73 | 0.95 | 0.29 | 0.98 | 0.45 | 0.60 | | 0.60 | 0.98 | 0.29 | 0.70 | 0.89 |
| General Merch.Stores | **0.37** | 0.28 | 0.06 | 0.57 | 0.73 | 0.80 | 0.88 | 0.52 | 0.77 | 0.36 | 0.50 | 0.60 | | 0.12 | 0.55 | 0.40 | 0.67 |
| Specialty Retail | **0.46** | 0.33 | 0.06 | 0.62 | 0.84 | 0.89 | 0.96 | 0.45 | 0.94 | 0.23 | 0.12 | 0.98 | 0.12 | | 0.46 | 0.52 | 0.84 |
| Electronics Retail | **0.46** | 0.41 | 0.22 | 0.57 | 0.62 | 0.21 | 0.15 | 0.30 | 0.63 | 0.93 | 0.54 | 0.29 | 0.55 | 0.46 | | 0.48 | 0.60 |
| Home Retail | 0.08 | 0.03 | 0.10 | 0.65 | 0.40 | 0.71 | 0.81 | 0.47 | 0.13 | 0.06 | 0.40 | 0.70 | 0.40 | 0.52 | 0.48 | | 0.59 |
| Textiles Goods | **0.46** | 0.27 | 0.19 | 0.53 | 0.24 | 0.87 | 0.91 | 0.58 | 0.46 | 0.24 | 0.67 | 0.89 | 0.67 | 0.84 | 0.60 | 0.59 | |

The table reports the p values of t-test checking the equality of means of AR(0) and AR(1) values. The null hypothesis states that the difference in group means is zero. An alternate hypothesis states that the difference in group means is different from zero. 1 refers to Auto Components; 2 Automobiles;3 Hotels; 4 Casinos and Gaming; 5 Restaurants; 6 Household Durables; 7 Home Building; 8 Household Appliances; 9 Consumer Electronics; 10 Internet and Direct Marketing Retail; 11 Multiline Retail; 12 Distributors; 13 General Merchandise Stores; 14 Specialty Retail; 15 Computers and Electronics Retail; 16 Home Furnishing Retail, 17 Textiles, Apparel and Luxury Goods. Data is obtained from Thomson Reuters Eikon for the period 23 June 2020 to 12 November 2020, for a total of 139 daily observations per index.

**Table 10. Consumer staples industry.**

| | Food&Staples Retailing | Drug Retail | Food Retail | Hypermarkets | Brewers | Soft Drinks | Food Prod. | Agricultural Prod. | Packaged Foods. | Household Prod. | Personal Prod. | Tobacco |
|---|---|---|---|---|---|---|---|---|---|---|---|---|
| Food and Staples Retailing | | 0.17 | 0.88 | 0.87 | 0.03 | 0.59 | 0.84 | 0.60 | 0.87 | 0.92 | 0.20 | 0.43 |
| Drug Retail | 0.17 | | 0.32 | 0.17 | 0.68 | 0.06 | 0.13 | 0.08 | 0.14 | 0.22 | 0.15 | 0.18 |
| Food Retail | 0.88 | 0.32 | | 0.89 | 0.44 | 0.60 | 0.64 | 0.57 | 0.64 | 0.78 | 0.67 | 0.41 |
| Hypermarkets and Super Centers | 0.87 | 0.17 | 0.89 | | 0.31 | 0.36 | 0.28 | 0.31 | 0.25 | 0.79 | 0.54 | 0.16 |
| Brewers | 0.03 | 0.68 | 0.44 | 0.31 | | 0.28 | 0.32 | 0.31 | 0.33 | 0.37 | 0.06 | 0.50 |
| Soft Drinks | 0.59 | 0.06 | 0.60 | 0.36 | 0.28 | | 0.47 | 0.66 | 0.48 | 0.49 | 0.88 | 0.17 |
| Food Products | 0.84 | 0.13 | 0.64 | 0.28 | 0.32 | 0.47 | | 0.36 | 0.55 | 0.50 | 0.71 | 0.08 |
| Agricultural Products | 0.60 | 0.07 | 0.57 | 0.31 | 0.31 | 0.66 | 0.36 | | 0.39 | 0.45 | 0.93 | 0.12 |
| Packaged Foods and Meats | 0.87 | 0.14 | 0.64 | 0.25 | 0.33 | 0.48 | 0.55 | 0.39 | | 0.50 | 0.70 | 0.10 |
| Household Products | 0.92 | 0.22 | 0.78 | 0.79 | 0.37 | 0.49 | 0.50 | 0.45 | 0.50 | | 0.62 | 0.26 |
| Personal Products | 0.20 | 0.15 | 0.67 | 0.54 | 0.06 | 0.77 | 0.71 | 0.93 | 0.70 | 0.62 | | 0.62 |
| Tobacco | 0.43 | 0.18 | 0.41 | 0.16 | 0.50 | 0.17 | 0.08 | 0.12 | 0.10 | 0.26 | 0.62 | |

The table reports the p values of t-test checking the equality of means of AR(0) and AR(1) values. The null hypothesis states that the difference in group means is zero. An alternate hypothesis states that the difference in group means is different from zero. Data is obtained from Thomson Reuters Eikon for the period 23 June 2020 to 12 November 2020, for a total of 139 daily observations per index.

appliances and computers and electronics retail. These results are in line with [33] who provide evidence that not all sectors comprising the Nasdaq-100 have reacted the same way.

These results suggest that even though the availability of vaccines is expected to help steer economies gradually back to normalcy, the re-adjustment is not going to be homogeneous for all subsectors. While some subsectors expect a recovery from the COVID-induced contraction, other subsectors face adjustment losses as these industries shed off the above average gains driven by the COVID-19 environment. These results are relevant for governments and

**Table 11. Industrials industry.**

| | Airlines | Railroads | Trans. Inf. | Air Freight&Log. | Building prod. | Aerospace | Elect.Eq. | Ind. Congl. | Machinery |
|---|---|---|---|---|---|---|---|---|---|
| Airlines | | 0.54 | 0.76 | 0.11 | 0.29 | 0.87 | 0.62 | 0.63 | 0.48 |
| Railroads | 0.54 | | 0.79 | 0.35 | 0.10 | 0.28 | 0.93 | 0.15 | 0.07 |
| Transportation Infrastructure | 0.76 | 0.79 | | 0.51 | 0.40 | 0.50 | 0.57 | 0.98 | 0.64 |
| Air Freight and Logistics | 0.11 | 0.35 | 0.51 | | 0.80 | 0.33 | 0.49 | 0.32 | 0.42 |
| Building products | 0.29 | 0.10 | 0.40 | 0.80 | | 0.18 | 0.33 | 0.11 | 0.14 |
| Aerospace and Defense | 0.87 | 0.28 | 0.50 | 0.33 | 0.18 | | 0.06 | 0.33 | 0.20 |
| Electrical Equipment | 0.62 | 0.93 | 0.57 | 0.49 | 0.33 | 0.06 | | 0.55 | 0.72 |
| Industrial Conglomerates | 0.63 | 0.15 | 0.98 | 0.32 | 0.11 | 0.33 | 0.55 | | 0.05 |
| Machinery | 0.48 | 0.07 | 0.64 | 0.42 | 0.14 | 0.20 | 0.72 | 0.05 | |

The table reports the p values of t-test checking the equality of means of AR(0) and AR(1) values. The null hypothesis states that the difference in group means is zero. An alternate hypothesis states that the difference in group means is different from zero. Data is obtained from Thomson Reuters Eikon for the period 23 June 2020 to 12 November 2020, for a total of 139 daily observations per index.

**Table 12. Information technology industry.**

|  | Comm. Equip. | IT Services | IT Consulting | Data Processing | Internet Serv. and Inf. | Semiconductors and Equip. | Software | Tech. Hardware and Storage |
|---|---|---|---|---|---|---|---|---|
| Communications Equipment |  | **0.89** | **0.38** | **0.99** | **0.12** | 0.03 | 0.05 |  |
| IT Services | **0.89** |  | **0.83** | **0.72** | **0.29** | **0.17** | **0.18** | **0.49** |
| IT Consulting and Other Services | **0.38** | **0.83** |  | **0.80** | 0.07 | 0.02 | 0.00 | **0.35** |
| Data Processing and Outsourcing | **0.99** | **0.72** | **0.80** |  | **0.34** | **0.22** | **0.23** | **0.51** |
| Internet Services and Infrastructure | **0.12** | **0.29** | 0.07 | **0.34** |  | **0.29** | **0.16** | **0.83** |
| Semiconductors and Equipment | 0.03 | **0.17** | 0.02 | **0.22** | **0.29** |  | **0.37** | **0.59** |
| Software | 0.05 | **0.18** | 0.00 | **0.23** | **0.16** | **0.37** |  | **0.51** |
| Technology Hardware and Storage | **0.35** | **0.49** | **0.35** | **0.51** | **0.83** | **0.59** | **0.51** |  |

The table reports the p values of t-test checking the equality of means of AR(0) and AR(1) values. The null hypothesis states that the difference in group means is zero. An alternate hypothesis states that the difference in group means is different from zero. Data is obtained from Thomson Reuters Eikon for the period 23 June 2020 to 12 November 2020, for a total of 139 daily observations per index.

**Table 13. Materials industry.**

|  | Chemicals | Construction Materials | Containers and Packaging | Metals and Mining |
|---|---|---|---|---|
| Chemicals |  | **0.86** | **0.48** | **0.16** |
| Construction Materials | **0.86** |  | **0.16** | **0.19** |
| Containers and Packaging | **0.48** | **0.16** |  | **0.20** |
| Metals and Mining | **0.16** | **0.19** | **0.20** |  |

The table reports the p values of t-test checking the equality of means of AR(0) and AR(1) values. The null hypothesis states that the difference in group means is zero. An alternate hypothesis states that the difference in group means is different from zero. Data is obtained from Thomson Reuters Eikon for the period 23 June 2020 to 12 November 2020, for a total of 139 daily observations per index.

**Table 14. Utilities industry.**

|  | Electric | Gas | Water | Multi |
|---|---|---|---|---|
| Electric |  | **0.49** | 0.04 | **0.85** |
| Gas | **0.49** |  | **0.37** | **0.36** |
| Water | 0.04 | **0.37** |  | **0.39** |
| Multi | **0.85** | **0.36** | **0.39** |  |

The table reports the p values of t-test checking the equality of means of AR(0) and AR(1) values. The null hypothesis states that the difference in group means is zero. An alternate hypothesis states that the difference in group means is different from zero. Data is obtained from Thomson Reuters Eikon for the period 23 June 2020 to 12 November 2020, for a total of 139 daily observations per index.

policymakers in managing the transition back to normalcy. Policy makers may have to provide support packages for industries expected to be negatively affected during the recovery phase. Furthermore, the results are relevant for portfolio managers. It suggests that portfolio diversification should consider the effects of external shocks such as pandemics, as sectors and sub-sectors are likely to be affected differently.

**Table 15. All industries.**

|  | Financials | Energy | Real Estate | Comm. Serv. | Healthcare | Cons. Discr. | Cons. Stapl. | Ind. | Inf. Tech. | Materials | Utilities |
|---|---|---|---|---|---|---|---|---|---|---|---|
| Financials |  | 0.09 | **0.78** | 0.03 | 0.09 | **0.33** | **0.14** | **0.14** | 0.03 | 0.07 | **0.12** |
| Energy | 0.09 |  | 0.30 | 0.00 | 0.00 | 0.03 | 0.01 | 0.01 | 0.00 | 0.00 | 0.00 |
| Real Estate | **0.78** | **0.20** |  | **0.35** | **0.23** | **0.47** | **0.35** | **0.52** | **0.50** | **0.53** | **0.42** |
| Comm. Serv. | 0.03 | 0.00 | **0.35** |  | **0.50** | **0.54** | **0.65** | 0.01 | **0.74** | **0.13** | **0.13** |
| Healthcare | 0.09 | 0.00 | **0.23** | **0.50** |  | **0.70** | **0.88** | **0.63** | **0.54** | **0.48** | **0.88** |
| Cons. Discr. | **0.33** | 0.03 | **0.47** | **0.54** | **0.70** |  | **0.59** | **0.49** | **0.86** | **0.17** | **0.25** |
| Cons. Stapl. | **0.14** | 0.01 | **0.35** | **0.65** | **0.88** | **0.59** |  | **0.34** | **0.69** | **0.66** | **0.28** |
| Industrials | **0.14** | 0.01 | **0.52** | 0.01 | **0.63** | **0.49** | **0.34** |  | **0.20** | **0.60** | **0.90** |
| Inf. Tech. | 0.03 | 0.00 | **0.50** | **0.74** | **0.54** | **0.86** | **0.69** | **0.20** |  | **0.17** | **0.79** |
| Materials | 0.07 | 0.00 | **0.53** | **0.13** | **0.48** | **0.17** | **0.66** | **0.60** | **0.17** |  | **0.26** |
| Utilities | **0.12** | 0.00 | **0.42** | **0.1** | **0.88** | **0.25** | **0.28** | **0.90** | **0.79** | **0.26** |  |

The table reports the p values of t-test checking the equality of means of AR(0) values. The null hypothesis states that the difference in group means is zero. An alternate hypothesis states that the difference in group means is different from zero. Data is obtained from Thomson Reuters Eikon for the period 23 June 2020 to 12 November 2020, for a total of 139 daily observations per index.

**Table 16. All industries.**

|  | Financials | Energy | Real Estate | Comm. Serv. | Healthcare | Cons. Discr. | Cons. Stapl. | Ind. | Inf. Tech. | Materials | Utilities |
|---|---|---|---|---|---|---|---|---|---|---|---|
| Financials |  | 0.03 | **0.46** | 0.02 | 0.09 | **0.96** | **0.35** | **0.59** | 0.03 | 0.05 | **0.12** |
| Energy | 0.03 |  | 0.08 | 0.00 | 0.00 | **0.50** | 0.03 | **0.25** | 0.00 | 0.00 | 0.00 |
| Real Estate | **0.46** | 0.08 |  | **0.47** | 0.09 | **0.24** | **0.71** | **0.54** | **0.70** | 0.00 | **0.87** |
| Comm. Serv. | 0.02 | 0.00 | **0.47** |  | **0.79** | **0.52** | **0.43** | **0.22** | **0.76** | **0.90** | **0.32** |
| Healthcare | 0.09 | 0.00 | 0.09 | **0.79** |  | **0.50** | **0.52** | **0.45** | **0.39** | **0.17** | **0.86** |
| Cons. Discr. | **0.96** | **0.50** | **0.24** | **0.52** | **0.50** |  | **0.50** | **0.55** | **0.57** | **0.18** | **0.36** |
| Cons. Stapl. | **0.35** | 0.02 | **0.71** | **0.43** | **0.52** | **0.50** |  | **0.64** | **0.22** | **0.34** | **0.81** |
| Industrials | **0.59** | **0.25** | **0.54** | **0.22** | **0.45** | **0.55** | **0.64** |  | **0.11** | **0.30** | **0.57** |
| Inf. Tech. | 0.03 | 0.00 | **0.70** | **0.76** | **0.39** | **0.57** | **0.22** | **0.11** |  | 0.06 | **0.93** |
| Materials | 0.05 | 0.00 | 0.00 | **0.90** | **0.17** | **0.18** | **0.34** | **0.30** | 0.06 |  | 0.04 |
| Utilities | **0.12** | 0.00 | **0.87** | **0.32** | **0.86** | **0.36** | **0.81** | **0.57** | **0.93** | 0.04 |  |

The table reports the p values of t-test checking the equality of means of CAR(0,3) values. The null hypothesis states that the difference in group means is zero. An alternate hypothesis staTes that the difference in group means is different from zero. Data is obtained from Thomson Reuters Eikon for the period 23 June 2020 to 12 November 2020, for a total of 139 daily observations per index.

As an extension of this study, future work should focus on examining the underlying factors that explain a considerable inter and intra sectoral differences in the impact of vaccine announcement. For this purpose, the sectoral determinants such as asset size, leverage and profitability ratios of the sectors/subsectors can be regressed on AR or CARs to estimate the likely determinants of different reactions.

Although studies [15, 44, 45] show that vaccination has a stabilising impact on the capital markets, a high vaccination coverage rate is necessary to reap full benefits of vaccination programs. However, vaccine hesitancy is considered a serious problem [46, 47] that needs to be addressed, especially in the developing countries [48]. More research is needed to devise optimal ways to counter vaccine hesitancy and to identify key factors promoting the slow uptake of COVID-19 vaccines in some communities and countries.

## Supporting information

**S1 Data.**
(XLSX)

## Author Contributions

**Conceptualization:** Burcu Kapar, Faisal Rana.

**Formal analysis:** Burcu Kapar.

**Methodology:** Burcu Kapar, Steven Buigut.

**Writing – original draft:** Burcu Kapar, Steven Buigut, Faisal Rana.

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
