## [Decision Letter · Decision Letter 0]

4 Jul 2022

PONE-D-22-14074Winners and Losers from Pfizer and Biontech's Vaccine Announcement: Evidence from S&P 500 (Sub)Sector IndicesPLOS ONE

Dear Dr. KAPAR,

Thank you for submitting your manuscript to PLOS ONE. After careful consideration, we feel that it has merit but does not fully meet PLOS ONE’s publication criteria as it currently stands. Therefore, we invite you to submit a revised version of the manuscript that addresses the points raised during the review process.

We look forward to receiving your revised manuscript.

Kind regards,

María del Carmen Valls Martínez, Ph.D.

Academic Editor

PLOS ONE

Journal Requirements:

3. Please ensure that you include a title page within your main document. You should list all authors and all affiliations as per our author instructions and clearly indicate the corresponding author.

Reviewers' comments:

Reviewer's Responses to Questions

**Comments to the Author**

1. Is the manuscript technically sound, and do the data support the conclusions?

Reviewer #1: Yes

Reviewer #2: Yes

Reviewer #3: Partly

2. Has the statistical analysis been performed appropriately and rigorously? 

Reviewer #1: Yes

Reviewer #2: Yes

Reviewer #3: Yes

3. Have the authors made all data underlying the findings in their manuscript fully available?

Reviewer #1: Yes

Reviewer #2: Yes

Reviewer #3: Yes

4. Is the manuscript presented in an intelligible fashion and written in standard English?

Reviewer #1: Yes

Reviewer #2: Yes

Reviewer #3: No

5. Review Comments to the Author

Reviewer #1: Dear Author/s

I would like to thank the author(s) for your submission and appreciate the opportunity to read and review your manuscript. I enjoyed reading it. My comments concern only with clarity, text, methodology used and interpretation of results. I would recommend publication, after the issues below are addressed.

Changes which must be made before publication:

Minor changes:

1. Abstract: please expand the abstract and add the methodology used.

2. The Keywords: the keywords does not accurately reflect the content of the study, additional 2-3 keywords should be added.

3. The introduction section: I think it is good.

4. The literature review section: please expand the literature review and add some latest references.

5. The methodology section: It's okay, but it could be better explained and motivated further. Please reference all numbered equations in text. Currently, numbered equations [1, 2, 3, 4] in the manuscript have not been cited in text.

6. The empirical results and discussion section: the results and discussion section should be better analyzed and developed further.

7. Conclusion section: It's okay, but it could be better explained and motivated with a few more minor adjustments, please try to augment its quality with more in-depth investigation and analysis:

a. I recommend to change the section title to (Conclusion and Policy Recommendations).

b. "limitations, and further research" could be added

8. The language of the paper: the language of the paper needs a careful editing by a native speaker.

Additional comments: the author/s are recommended to do the following to increase the paper readability.

1. To start with, the paper should more clearly and more explicitly spell out its objectives.

2. Develop the literature review section of the article to include 3-5 latest journal references (2021-2022) and relevant extracts from them.

3. Please, avoid placing tables or figures before their first mention in the text, and the analysis should always be below the figure or the table.

4. Please avoid the following word/s; "our analysis" (please see p. 5), "our results" (please see p.8) , "our earlier results" (please see p.9). Please replace with "the present study………" or "the current study………"…etc.

5. Please be careful in using the abbreviations and abbreviated words throughout the paper.

6. Please be careful in using "Punctuating", e.g. (comma, full stop, etc…) throughout the paper.

Reviewer #2: Recommendations for Manuscript PONE-D-22-14074 „Winners and Losers from Pfizer and Biontech's Vaccine Announcement: Evidence from S&P 500 (Sub)Sector Indices” for the PLOS ONE Journal.

General Comments

From my point of view, it is a very interesting topic and simultaneously it seems that to the best of my knowledge is the first empirical research which explores how various sectors and subsectors of the US stock market react to the news of successful development of vaccine by Pfizer and Biontech on November 9, 2020. Based on eleven sector and eighty subsector indices of S&P 500, the authors establish that there are considerable inter and intra sectoral variations in the impact of the vaccine news, and that the impact follows a clear pattern.

The paper contains the following sections: Introduction, Literature Review, Data, Methodology, Empirical Results and Discussion and Conclusion.

However, I find some recommendations:

1. The abstract must contain the main purpose of the paper, the research method used in the research and the main contributions.

2. It would be very useful to add in the "Introduction" section the purpose, objectives and hypothesis of the research. I consider that a weak point of the paper is that the authors did not show the novelty of the paper compared to other works. That is why, I consider that the introduction should specify the novelty of the paper compared to other papers published in this area.

3. The research is well based on science and the results are in agreement with the theoretical part. The model applied to the analyzed data is correctly used in the analysis undertaken, it is a strength point of this paper.

4. At the same time, the authors are required to present Descriptive Statistics, Correlation matrix with all tests and indicators: standard deviation, Jarqe-Berra, Skewness and Kurtosis interpretation, Jarqe-Berra with probabilities analysis, etc.

5. It is important to present the VIF test on multicollinearity between independent variables. Heteroskedasticity and endogeneity tests are also important in this study. All these aspects that are not found in the paper represent weaknesses of the research.

6. I think that the literature needs to be improved with other recent works, refers to the companies listed on the stock market. That is why I recommend the authors to refer to other recent works indexed in Web of Science, Scopus, Emerald and Cambrige Journals. We suggest that the authors cite papers published in Web of Science Journals, such as:

1. Batrancea, L.; Rus, M.I.; Masca, E.S.; Morar, I.D. Fiscal Pressure as a Trigger of Financial Performance for the Energy Industry: An Empirical Investigation across a 16-Year Period. Energies 2021, 14, 3769. https://doi.org/10.3390/en14133769

2. Batrancea, L., Rathnaswamy, M.K. & Batrancea, I. A Panel Data Analysis on Determinants of Economic Growth in Seven Non-BCBS Countries. J Knowl Econ (2021). https://doi.org/10.1007/s13132-021-00785-y

3. Batrancea, L. (2021) An Econometric Approach Regarding the Impact of Fiscal Pressure on Equilibrium: Evidence from Electricity, Gas and Oil Companies Listed on the New York Stock Exchange, Mathematics 9, no. 6: 630. https://doi.org/10.3390/math9060630.ISSN:2227-7390

7. Based on the data obtained, the conclusions must be extended.

In conclusion, the article should be improve. It should also be enhanced with a review of the literature adequate to the subject and a broader interpretation and commentary of the research results.

Reviewer #3: The authors are able to provide his readers a solid, clear and adequate justification of the problem, the authors' should need a keen review of the whole paper. Paper is good and can be considered after making improvements described in review report.

6. PLOS authors have the option to publish the peer review history of their article (what does this mean?). If published, this will include your full peer review and any attached files.

Reviewer #1: No

Reviewer #2: No

Reviewer #3: No

---

## [Author Response · Author response to Decision Letter 0]

22 Aug 2022

Answer to the Referee Report – PLOS One

August 17, 2022

First of all, we would like to express our gratitude for your careful reading and comments on

our manuscript. They have helped to clarify important aspects of the manuscript that were unclear in the previous version. We worked on these comments and believe our work benefited as a result. In this document, we provide our response to each of the points raised in your referee report.

Reviewer 1

I would like to thank the author(s) for your submission and appreciate the opportunity to read and review your manuscript. I enjoyed reading it. My comments concern only with clarity, text, methodology used and interpretation of results. I would recommend publication, after the issues below are addressed.

Changes which must be made before publication: 

Minor changes:

1. Abstract: please expand the abstract and add the methodology used. 

Answer: Abstract has been rewritten including the methodology used.

2. The Keywords: the keywords does not accurately reflect the content of the study, additional 2-3 keywords should be added.

Answer: We have added COVID-19 pandemic, Pfizer and Biontech vaccine, Event Study and Overreaction.

3. The introduction section: I think it is good.

4. The literature review section: please expand the literature review and add some latest references.

Answer: This section has been reviewed and recently published papers added.

5. The methodology section: It's okay, but it could be better explained and motivated further. Please reference all numbered equations in text. Currently, numbered equations [1, 2, 3, 4] in the manuscript have not been cited in text. 

Answer: We referenced all numbered equations in the text.

6. The empirical results and discussion section: the results and discussion section should be better analyzed and developed further.

Answer: Discussion section has been rewritten including the robustness section.

7. Conclusion section: It's okay, but it could be better explained and motivated with a few more minor adjustments, please try to augment its quality with more in-depth investigation and analysis:

a. I recommend to change the section title to (Conclusion and Policy Recommendations).

b. "limitations, and further research" could be added

Answer: Title is changed as recommended. Limitations and Further research has been improved.

8. The language of the paper: the language of the paper needs a careful editing by a native speaker.

Answer: Paper is proofread.

Additional comments: the author/s are recommended to do the following to increase the paper readability.

1. To start with, the paper should more clearly and more explicitly spell out its objectives. 

Answer: Objective section is rewritten. 

2. Develop the literature review section of the article to include 3-5 latest journal references (2021-2022) and relevant extracts from them.

Answer: Recently published papers about the topic are added.

3. Please, avoid placing tables or figures before their first mention in the text, and the analysis should always be below the figure or the table.

Answer: Tables and figures are placed after they are mentioned. As there are many tables and some in the landscape format, majority of the tables are placed at the end of the paper as “Supporting Information.”

4. Please avoid the following word/s; "our analysis" (please see p. 5), "our results" (please see p.8) , "our earlier results" (please see p.9). Please replace with "the present study………" or "the current study………"…etc.

Answer: All changes are made.

5. Please be careful in using the abbreviations and abbreviated words throughout the paper.

Answer: Abbreviations are removed. Only AR(Abnormal Return) and CAR (Cumulative Abnormal Return) are used throughout the paper.

6. Please be careful in using "Punctuating", e.g. (comma, full stop, etc…) throughout the paper.

Answer: Paper is proofread.

Reviewer 2

From my point of view, it is a very interesting topic and simultaneously it seems that to the best of my knowledge is the first empirical research which explores how various sectors and subsectors of the US stock market react to the news of successful development of vaccine by Pfizer and Biontech on November 9, 2020. Based on eleven sector and eighty subsector indices of S&P 500, the authors establish that there are considerable inter and intra sectoral variations in the impact of the vaccine news, and that the impact follows a clear pattern.

The paper contains the following sections: Introduction, Literature Review, Data, Methodology, Empirical Results and Discussion and Conclusion.

However, I find some recommendations:

1. The abstract must contain the main purpose of the paper, the research method used in the research and the main contributions.

Answer: Abstract has been rewritten. 

2. It would be very useful to add in the "Introduction" section the purpose, objectives and hypothesis of the research. I consider that a weak point of the paper is that the authors did not show the novelty of the paper compared to other works. That is why, I consider that the introduction should specify the novelty of the paper compared to other papers published in this area.

Answer: Introduction section has been rewritten including the points mentioned.

3. The research is well based on science and the results are in agreement with the theoretical part. The model applied to the analyzed data is correctly used in the analysis undertaken, it is a strength point of this paper.

4. At the same time, the authors are required to present Descriptive Statistics, Correlation matrix with all tests and indicators: standard deviation, Jarqe-Berra, Skewness and Kurtosis interpretation, Jarqe-Berra with probabilities analysis, etc.

Answer: Table 1 reports descriptive statistics (mean, standard deviation, minimum, maximum, skewness, kurtosis) and Jarque Berra Test statistics. In the Data section, there is interpretation of the statistics. Supporting information document is submitted separately reporting the Pearson Correlation Coefficients (requested) among main industries, and sub-sectors under each industry. However, note that our analysis is carried out using the market model and the computation of the abnormal returns is done sector (subsector) by sector (subsector). The correlation matrix of sectors (subsectors) does not therefore provide much addition information. 

5. It is important to present the VIF test on multicollinearity between independent variables. Heteroskedasticity and endogeneity tests are also important in this study. All these aspects that are not found in the paper represent weaknesses of the research.

Answer: 

On VIF: The event study analysis approach (market model) that we use involves the use of one independent variable (market index) at a time. Hence there is no element of multicollinearity. We do not therefore need to carry out VIF. 

On endogeneity, our primary (and only) independent variable is market index. To test for endogeneity requires a good IV. As most literature generally show, it is not easy to get a good IV, and in particular it is a challenge to get a good instrument for market index. It is part of the reason event studies (market model) do not generally provide tests for endogeneirty. For this reason we do not provide endogeneity tests. 

To check the heteroskedasticity, we have applied the Breusch-Pagan test on the residuals from our market model. The table below shows the p value of some of the tests. The null hypothesis of the tests states that there is generally a constant variance (homescedasticity). This indicates that generally there is no heteroskedasticity in market model regression and confirms the findings of our methodology.

Dependant Variable Independent Variable P value

Communication Services Index Market Index 0.1678

Utilities Index Market Index 0.0781

Energy Index Market Index 0.1055

Financials Index Market Index 0.0835

Real Estate Index Market Index 0.9275

Consumer Staples Index Market Index 0.2561

Consumer Discretionary Index Market Index 0.1785

Industrials Index Market Index 0.3066

Information Technology Index Market Index 0.4503

6. I think that the literature needs to be improved with other recent works, refers to the companies listed on the stock market. That is why I recommend the authors to refer to other recent works indexed in Web of Science, Scopus, Emerald and Cambrige Journals. We suggest that the authors cite papers published in Web of Science Journals, such as:

1. Batrancea, L.; Rus, M.I.; Masca, E.S.; Morar, I.D. Fiscal Pressure as a Trigger of Financial Performance for the Energy Industry: An Empirical Investigation across a 16-Year Period. Energies 2021, 14, 3769. https://doi.org/10.3390/en14133769

2. Batrancea, L., Rathnaswamy, M.K. & Batrancea, I. A Panel Data Analysis on Determinants of Economic Growth in Seven Non-BCBS Countries. J Knowl Econ (2021). https://doi.org/10.1007/s13132-021-00785-y

3. Batrancea, L. (2021) An Econometric Approach Regarding the Impact of Fiscal Pressure on Equilibrium: Evidence from Electricity, Gas and Oil Companies Listed on the New York Stock Exchange, Mathematics 9, no. 6: 630. https://doi.org/10.3390/math9060630.ISSN:2227-7390

Answer: The references mentioned have been added. 

7. Based on the data obtained, the conclusions must be extended.

In conclusion, the article should be improve. It should also be enhanced with a review of the literature adequate to the subject and a broader interpretation and commentary of the research results.

Answer: Conclusion section has been rewritten and limitations of the research has been added.

Reviewer 3

In the underlined research topic “Winners and Losers from Pfizer and Biontech's Vaccine Announcement: Evidence from S&P 500 (Sub)Sector Indices” is a novel part to the existing literature. The title of the article is remarkable, the presentation is good and the article may provide several scientific justifications. Below are several suggestions which will increase writer input to technical research and probability.

1. The paper needs to be improved in all parts (justification of the contribution, conceptual background, the method). I hope you find the reviewers' comments helpful in developing the paper further.

2. A single paragraph serves as the article's assessment of the literature, and the commentary is disorganized and illogical. I'd advise the writers to restructure the literature review section and make the logical connections clearer.

Answer: This part has been rewritten.

3. Globally, the manuscript needs a revision of the English and there are paragraphs and sentences written in a very confusing way.

Answer: Paper is proofread.

4. The technique used in the study is appropriate for the findings, and but the authors should review it again to make is more attractive.

Answer: This section has been improved.

5. At the bottom of the table, provide a note identifying the data source.

Answer: The data source is mentioned under each table.

6. The author should expand on the empirical findings.

Answer: Empirical parts have been rewritten.

7.To make the paper statistically strong, the authors should need to add more innovative techniques to make the results much clear.

Answer: This section has been improved.

8.The results follow the methodology but the interpretation of tables and graphs need more quality presentation.

Answer: Tables have been improved.

9.The results and discussion of the study is quite lengthy despite the importance of the information but if possible to eliminate some of the secondary information that does not disturb the quality of the research flow.

Answer: Discussion section is improved.

10. A separate set of data is needed for the author.

Answer: Data set will be submitted to the journal.

11.To write an equation, the author must adhere to a prescribed format.

Answer: Equations are rewritten and referenced in the text.

12."Illustration from simulation" is the desired outcome of the simulation research.

Answer: This is removed from the paper.

13. The author must distinguish the inference from the conclusion in order to avoid confusion.

Answer: This is taken into account in the discussion and conclusion sections.

14. Referencing should follow a standard structure that is consistent with the references mentioned in journal authors’ guidelines.

Answer: References section is updated in line with the journal requirement.

15. Discussions should add some latest references of 2021 and 2022 to prove the results.

Answer: Latest references are added in Introduction and Discussion sections.

16. More explicitly stated restrictions and future proposals should be included in this article.

Answer: Limitations and further research are added to the Conclusion.

Finally, it’s a nice attempt, the topic is of interest and has a wide range of policy implementations. Bases on the above-mentioned comments, I suggest “major revision” of the article for more practicability before “publication”.

---

## [Decision Letter · Decision Letter 1]

6 Sep 2022

PONE-D-22-14074R1Winners and Losers from Pfizer and Biontech's Vaccine Announcement: Evidence from S&P 500 (Sub)Sector IndicesPLOS ONE

Dear Dr. Kapar,

Thank you for submitting your manuscript to PLOS ONE. After careful consideration, we feel that it has merit but does not fully meet PLOS ONE’s publication criteria as it currently stands. Therefore, we invite you to submit a revised version of the manuscript that addresses the points raised during the review process.

We look forward to receiving your revised manuscript.

Kind regards,

María del Carmen Valls Martínez, Ph.D.

Academic Editor

PLOS ONE

Journal Requirements:

Reviewers' comments:

Reviewer's Responses to Questions

**Comments to the Author**

1. If the authors have adequately addressed your comments raised in a previous round of review and you feel that this manuscript is now acceptable for publication, you may indicate that here to bypass the “Comments to the Author” section, enter your conflict of interest statement in the “Confidential to Editor” section, and submit your "Accept" recommendation.

Reviewer #1: All comments have been addressed

Reviewer #2: All comments have been addressed

2. Is the manuscript technically sound, and do the data support the conclusions?

Reviewer #1: Yes

Reviewer #2: Yes

3. Has the statistical analysis been performed appropriately and rigorously? 

Reviewer #1: Yes

Reviewer #2: Yes

4. Have the authors made all data underlying the findings in their manuscript fully available?

Reviewer #1: Yes

Reviewer #2: Yes

5. Is the manuscript presented in an intelligible fashion and written in standard English?

Reviewer #1: Yes

Reviewer #2: Yes

6. Review Comments to the Author

Reviewer #1: Dear Author(s)

I would like to thank the author(s) for addressing my initial comments. The author(s) have greatly improved their manuscript and responded to the points that I have raised. Following the revision to the paper, some of my additional "minor comments" relate to some of the amendments made. My remaining comments concern only with clarity, text, and methodology. I would recommend publication, after the issues below are addressed.

Minor changes:

1. The introduction section: Some of the terminology used in the manuscript needs to be unified. For instance, the use of two different words that have the same meaning interchangeably could confuse some readers. Dear author(s)please see the introduction section, p. 3 line 16-17 the author(s) mentioned that " "Section 5 presents empirical findings and discussion, and Section 6 concludes", while in p. 9 line 9 the author(s)used another words "5 Empirical results and Discussion", then in p. 14 line 1 the author(s)used another words "6 Conclusion and Policy Recommendations".

2. The Literature review section: In the first paragraph the author(s) mentioned that "With the outbreak of the coronavirus (COVID-19) pandemic, an increasing number of researchers have examined the impact of the pandemic on international financial markets. Towards this end, recent studies have established that the pandemic impacted stock markets (Acharya et al., 2021; Al-Awadhi et al., 2020; Baek et al., 2020; Engelhardt et al., 2021; Kapar et al., 2021; Kucher et al., 20221; Rouatbi et al., 2021), bond markets (Augustin et al., 2022; Chen et al., 2021; Falato et al., 2021; Haddad et al., 2021), gold markets (Corbert et al., 2020; Gharib et al. 2021; Mensi et al., 2020), exchange rate (Li et al., 2021; Aquilante et al., 2022; Jamal and Bhat, 2022), crude oil markets (Sharif et al., 2020; Mensi et al., 2020; Shaikh, 2021; Gharib et al., 2021; Wang et al., 2022), cryptocurrency markets (Corbert et al., 2020; Conlon and McGee, 2020; Yarovaya et al., 2020; Vidal-Tomás, 2021; Hong and Yoon, 2022), real estate markets (Balemi et al., 2021; Ling et al., 2020; Milcheva et al., 2022), and other asset classes". It contains too a large number of references, I would suggest you devote some effort to discuss such previous studies and – more importantly – their relationships and the major factors responsible for differences in results from same study conducted by different researchers.

3. The methodology section: The description of the methods is much clearer now. But I would suggest the author(s)to add a new subtitle after the methodology section title. Please add a subtitle (4.1 Event Study Methodology) then add a single paragraph discussing the theoretical basis of this methodology to help the reader to follow the section better.

Finally, based on the above-mentioned comments, I suggest “minor revision” of the paper before publication.

Reviewer #2: The authors integrated the recommendations made by the reviewer into the paper. I agree with the publication of this paper in the prestigious journal PLOS ONE.

7. PLOS authors have the option to publish the peer review history of their article (what does this mean?). If published, this will include your full peer review and any attached files.

Reviewer #1: No

Reviewer #2: No

---

## [Author Response · Author response to Decision Letter 1]

17 Sep 2022

September 7, 2022

First of all, we would like to express our gratitude for your careful reading and comments on

our manuscript. They have helped to clarify important aspects of the manuscript that were unclear in the previous version. We worked on these comments and believe our work benefited as a result. In this document, we provide our response to each of the points raised in your referee report.

Minor changes:

1. The introduction section: Some of the terminology used in the manuscript needs to be unified. For instance, the use of two different words that have the same meaning interchangeably could confuse some readers. Dear author(s)please see the introduction section, p. 3 line 16-17 the author(s) mentioned that " "Section 5 presents empirical findings and discussion, and Section 6 concludes", while in p. 9 line 9 the author(s)used another words "5 Empirical results and Discussion", then in p. 14 line 1 the author(s)used another words "6 Conclusion and Policy Recommendations".

Answer: This is corrected. In Page 3, we mentioned as this: Section 2 discusses the literature review; Section 3 presents the data; Section 4 explains the methodology; Section 5 presents empirical results and discussion; and Section 6 concludes and presents policy recommendations.

2. The Literature review section: In the first paragraph the author(s) mentioned that "With the outbreak of the coronavirus (COVID-19) pandemic, an increasing number of researchers have examined the impact of the pandemic on international financial markets. Towards this end, recent studies have established that the pandemic impacted stock markets (Acharya et al., 2021; Al-Awadhi et al., 2020; Baek et al., 2020; Engelhardt et al., 2021; Kapar et al., 2021; Kucher et al., 20221; Rouatbi et al., 2021), bond markets (Augustin et al., 2022; Chen et al., 2021; Falato et al., 2021; Haddad et al., 2021), gold markets (Corbert et al., 2020; Gharib et al. 2021; Mensi et al., 2020), exchange rate (Li et al., 2021; Aquilante et al., 2022; Jamal and Bhat, 2022), crude oil markets (Sharif et al., 2020; Mensi et al., 2020; Shaikh, 2021; Gharib et al., 2021; Wang et al., 2022), cryptocurrency markets (Corbert et al., 2020; Conlon and McGee, 2020; Yarovaya et al., 2020; Vidal-Tomás, 2021; Hong and Yoon, 2022), real estate markets (Balemi et al., 2021; Ling et al., 2020; Milcheva et al., 2022), and other asset classes". It contains too a large number of references, I would suggest you devote some effort to discuss such previous studies and – more importantly – their relationships and the major factors responsible for differences in results from same study conducted by different researchers.

Answer: We have updated this paragraph and discuss the papers that talks about the effect of COVID-19 only on stock markets in details as our paper is related to the stock market. The additional two paragraphs are as below:

With the outbreak of the coronavirus (COVID-19) pandemic, an increasing number of researchers have examined the impact of the pandemic on stock markets. Baker et al. (2020) document that no previous infectious disease outbreak, including the Spanish Flu, has impacted the stock market as forcefully as the COVID-19 pandemic due to strict government restrictions on commercial activity and voluntary social distancing. Exploring the direct effects and spillovers of COVID-19, He et al. (2020a) find that COVID-19 has a negative but short-term impact on stock markets of affected countries. By using a large sample of 63 stock markets covering all key markets, Kapar et al. (2021) find that the Wuhan lockdown induces negative spillover effects on markets in Europe, North America and other global markets. This is mainly attributed to fear and uncertainty as these markets had yet to introduce domestic restrictions and had minimal infections at the time. The rapid transmission of cases outside China particularly in Europe and the introduction of containment measures result in severe market decline which highlights the need for quick, globally coordinated response to contagious diseases. 

Controlling for traditional market drivers (such as investor sentiment, credit risk, liquidity risk, safe-haven asset demand and the price of oil), O’Donnell et al. (2021) conclude that the daily total count of confirmed COVID-19 cases is a leading factor in influencing equity prices. Using panel data analysis, Al-Awadhi et al. (2020) estimate that both the daily growth in confirmed cases and number of deaths caused by COVID-19 have significant negative effects on returns in Chinese stock market. Ashraf (2020) finds that stock markets react more strongly to the growth in number of confirmed cases as compared to the growth in number of deaths. Mazur et al. (2021) examine the US stock market during the crash of March 2020. They estimate that approximately 90% of the S&P 1500 stocks generate asymmetrically distributed large negative returns. The consensus from this emerging literature on COVID-19 suggests that stock markets respond negatively and significantly to COVID-19. Individual stock responses may vary, however, depending on several factors.

3. The methodology section: The description of the methods is much clearer now. But I would suggest the author(s)to add a new subtitle after the methodology section title. Please add a subtitle (4.1 Event Study Methodology) then add a single paragraph discussing the theoretical basis of this methodology to help the reader to follow the section better.

Answer: 4.1 Event Study Methodology subtitle is included under Methodology section. A paragraph is added to discuss the theoretical basis of this methodology.

---

## [Editor Report · Decision Letter 2]

26 Sep 2022

Winners and Losers from Pfizer and Biontech's Vaccine Announcement: Evidence from S&P 500 (Sub)Sector Indices

PONE-D-22-14074R2

Dear Dr. Burcu Kapar,

We’re pleased to inform you that your manuscript has been judged scientifically suitable for publication and will be formally accepted for publication once it meets all outstanding technical requirements.

Kind regards,

María del Carmen Valls Martínez, Ph.D.

Academic Editor

PLOS ONE
---

## [Editor Report · Acceptance letter]

6 Oct 2022

PONE-D-22-14074R2 

Winners and Losers from Pfizer and Biontech's Vaccine Announcement: Evidence from S&P 500 (Sub)Sector Indices 

Dear Dr. Kapar:

I'm pleased to inform you that your manuscript has been deemed suitable for publication in PLOS ONE. Congratulations! Your manuscript is now with our production department. 

Kind regards, 

on behalf of

Dr. María del Carmen Valls Martínez 

Academic Editor

PLOS ONE